# Engineering new limits to magnetostriction through metastability in iron-gallium alloys

P. B. Meisenheimer [1], R. A. Steinhardt[2], S. H. Sung [1], L. D. Williams [3], S. Zhuang[4], M. E. Nowakowski[5], S. Novakov [6], M. M. Torunbalci [7], B. Prasad[8], C. J. Zollner[9], Z. Wang[9], N. M. Dawley [2], J. Schubert [10], A. H. Hunter[11], S. Manipatruni[12], D. E. Nikonov [12], I. A. Young [12], L. Q. Chen [13], J. Bokor [5], S. A. Bhave [7], R. Ramesh [8,14,15], J.-M. Hu [4], E. Kioupakis[1], R. Hovden [1], D. G. Schlom [2,16,17] & J. T. Heron [1✉]

Magnetostrictive materials transduce magnetic and mechanical energies and when combined with piezoelectric elements, evoke magnetoelectric transduction for high-sensitivity magnetic field sensors and energy-efficient beyond-CMOS technologies. The dearth of ductile, rare-earth-free materials with high magnetostrictive coefficients motivates the discovery of superior materials. $Fe_{1-x}Ga_x$ alloys are amongst the highest performing rare-earth-free magnetostrictive materials; however, magnetostriction becomes sharply suppressed beyond $x = 19\%$ due to the formation of a parasitic ordered intermetallic phase. Here, we harness epitaxy to extend the stability of the BCC $Fe_{1-x}Ga_x$ alloy to gallium compositions as high as $x = 30\%$ and in so doing dramatically boost the magnetostriction by as much as 10x relative to the bulk and 2x larger than canonical rare-earth based magnetostrictors. A $Fe_{1-x}Ga_x - [Pb(Mg_{1/3}Nb_{2/3})O_3]_{0.7}-[PbTiO_3]_{0.3}$ (PMN-PT) composite magnetoelectric shows robust 90° electrical switching of magnetic anisotropy and a converse magnetoelectric coefficient of $2.0 \times 10^{-5}\, s\, m^{-1}$. When optimally scaled, this high coefficient implies stable switching at ~80 aJ per bit.

[1] Department of Materials Science and Engineering, University of Michigan, Ann Arbor, MI, USA. [2] Department of Materials Science and Engineering, Cornell University, Ithaca, NY, USA. [3] Department of Materials Design and Innovation, University at Buffalo - The State University of New York, Buffalo, NY, USA. [4] Department of Materials Science and Engineering, University of Wisconsin-Madison, Madison, WI, USA. [5] Department of Electrical Engineering and Computer Sciences, University of California, Berkeley, CA, USA. [6] Department of Physics, University of Michigan, Ann Arbor, MI, USA. [7] OxideMEMS Lab, Purdue University, West Lafayette, IN, USA. [8] Department of Materials Science and Engineering, University of California, Berkeley, CA, USA. [9] School of Applied and Engineering Physics, Cornell University, Ithaca, NY, USA. [10] Peter Grünberg Institute (PGI-9) and JARA Fundamentals of Future Information Technology, Forschungszentrum Jülich GmbH, Jülich, Germany. [11] Michigan Center for Materials Characterization, University of Michigan, Ann Arbor, MI, USA. [12] Components Research, Intel Corporation, Hillsboro, OR, USA. [13] Department of Materials Science and Engineering, Penn State University, State College, PA, USA. [14] Materials Sciences Division, Lawrence Berkeley National Laboratory, CA, USA. [15] Department of Physics, University of California, Berkeley, CA, USA. [16] Kavli Institute at Cornell for Nanoscale Science, Ithaca, NY, USA. [17] Leibniz-Institut für Kristallzüchtung, Max-Born-Str. 2, Berlin, Germany. ✉email: jtheron@umich.edu

Magnetostriction, the coupling of strain and magnetic order in materials, is a key parameter in the function of composite magnetoelectric multiferroic systems[1–3]. Such systems are desirable for applications in low-energy, beyond-CMOS technologies[4,5], and extremely sensitive magnetic field sensors[6,7]. Composite materials where a magnetostrictor is coupled to a piezoelectric crystal through intermediate strain, enabling electrical control of magnetization, offer increased magnetoelectric coefficients and device utility[8,9], when compared to rare, single-phase multiferroics[10–12], by combining the advantages of a wide array of magnetostrictive ferromagnets (Ni, $Fe_{1-x}Ga_x$, Terfenol-D, etc.) and piezoelectric substrates ($PbZr_{1-x}Ti_xO_3$, PMN-PT, etc.). While existing composites show impressive electrically driven magnetic reorientation capabilities, device performance has been limited by the magnetostrictive properties of the magnetic layer. It has recently been noted that a significant need in the field is the engineering of magnetostrictive magnets[13], as existing material systems have small coefficients, such as Ni or CoFeB are expensive and difficult to process, as in the case of rare-earth-based magnetic alloys (Terfenol-D).

The transition metal alloy $Fe_{1-x}Ga_x$ is an exciting candidate for an efficient, rare-earth-free magnetostrictor due to the large spin-orbit coupling and unique electronic structure. Recent observations show that this leads to a significant, compositionally dependent anomalous Nernst effect[14,15] and strongly lattice tunable magnetocrystalline anisotropy[16–18] in a rare-earth free system. A significant challenge, however, is that bulk, $Fe_{1-x}Ga_x$ undergoes a phase transition at ~19% Ga from a disordered A2 phase, at low concentrations of Ga, to an ordered BCC-like phase ($D0_3$)[17], capping the magnetostriction of the alloy at values around 300 ppm[18]. Until this phase change, the magnetostriction of $Fe_{1-x}Ga_x$ increases with increasing Ga incorporation, but sharply drops after 19% due to the formation of the parasitic intermetallic phase[19]. Previous reports have shown that the formation of this $D0_3$ phase can be suppressed by quenching, extending the usable range of the A2 phase to about 19% Ga and increasing the peak magnetostriction coefficient[20]. Hypothesizing that a similar effect could be achieved in thin films by employing a substrate stabilizing the desired polymorph via epitaxy[21], we push deep into the metastable range and synthesize the chemically disordered BCC (A2) phase in our films at gallium concentrations up to 30%. Furthermore, we leverage the second phase transition known in bulk to occur in $Fe_{1-x}Ga_x$ at ~30% Ga and the accompanying lattice softening to further increase the magnetostriction. Here, we investigate thin films of the magnetostrictive alloy $Fe_{1-x}Ga_x$ due to its earth abundance, high magnetostriction, and relatively unexplored phase space.

In the thin film regime, direct measurement of mechanical properties, such as magnetostriction, is difficult and remains an active area of research. State-of-the-art techniques can evaluate elastic moduli of simple systems with thicknesses on the order of multiple 10 s of nm, often by borrowing assumptions from bulk materials[22–25]. In magnetics, magnetostriction of thin films may be extracted indirectly from their magnetoelastic coupling coefficients[26] by analyzing the strain-induced change in magnetic anisotropy[27,28]. Our approach takes direction from these existing methodologies[26–28], but leverages strain from an intrinsic piezoelectric component. By evaluating the magnetostriction of $Fe_{1-x}Ga_x$ through magnetoelectric measurements and theoretical analysis based on piezoelectric domain imaging, we present a means to boost the magnetostriction by as much as 20× through epitaxial engineering and utilize this to demonstrate bipolar, 90° switching of magnetization via an electric field in a device with exceptional performance. Our results demonstrate a route to tap the spectacular properties that metastable polymorphs of common materials can provide; in this case yielding magnetostrictors and magnetoelectric multiferroics with unparalleled performance.

## Results

**Structure.** Our samples consist of epitaxial (001)-oriented, 15 nm thick A2 ($\alpha$-Fe) phase, magnetostrictive $Fe_{1-x}Ga_x$ ($x = 0.215$, 0.245, 0.30) single-crystal films on 1 mm-thick (001)-oriented PMN-PT substrates deposited by molecular-beam epitaxy (Fig. 1a). High-angle annular dark-field scanning transmission electron (HAADF-STEM) micrographs (Fig. 1b) reveal the deposition of single crystalline, phase-pure $Fe_{1-x}Ga_x$ films on

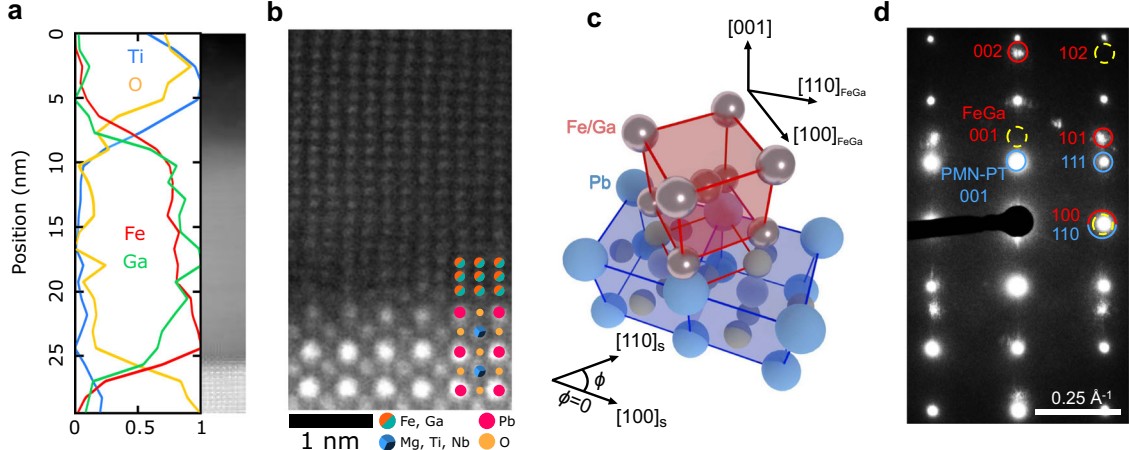

**Fig. 1 Epitaxial stabilization of A2 $Fe_{1-x}Ga_x$ on (001) PMN-PT. a** Electron energy loss spectroscopy (EELS) of the PMN-PT substrate and $Fe_{1-x}Ga_x$ film as a function of film thickness, showing abrupt concentration edges and a nominal thickness of ~15 nm for the $Fe_{1-x}Ga_x$ film. Ti signal comes from the capping layer to prevent oxidation. **b** High-angle annular dark-field scanning transmission electron micrographs (HAADF-STEM) along the PMN-PT [100] / $Fe_{1-x}Ga_x$ [110] zone axis, showing the single crystalline, epitaxial relationship along the $[100]_s$ substrate direction. **c** Diagram showing the epitaxial relationship of PMN-PT (blue) and $Fe_{1-x}Ga_x$ (red) normal to the interface ([001] direction); crystallographic directions of the film and the substrate are shown. **d** Interfacial selected area electron diffraction (SAED) confirming that the $Fe_{1-x}Ga_x$ thin film is in the disordered A2 phase due to the absence of superlattice peaks which would appear in the ordered $D0_3$ phase. Bragg peaks of the $Fe_{1-x}Ga_x$ (red) only appear when the sum of reciprocal lattice indices is even (missing peaks shown as yellow dashed circles), indicating a solid solution BCC crystal structure. Parts (**a**), (**b**), and (**d**) are collected from a representative 30% Ga sample, full diffraction data are shown in Sup. Fig. S1.

PMN-PT with a clean and coherent interface. We observe the epitaxial orientation relationship $[100]_s$ PMN-PT//$[110]$ Fe$_{1-x}$Ga$_x$ with a 45° in-plane rotation between Fe$_{1-x}$Ga$_x$ and PMN-PT from a cube-on-cube orientation relationship. Crystallographic directions in PMN-PT are referred to in their pseudocubic indices for clarity. This relationship is shown schematically in Fig. 1c. Selected area electron diffraction (SAED) images show only peaks with even Bragg indices (Fig. 1d), indicating the films are in a disordered body-centered cubic (BCC) crystal structure ($\alpha$-Fe phase) due to the absence of ordered superlattice peaks. The relative Ga: Fe concentration in our samples was measured to be 21.5 ± 3%Ga using quantitative electron energy loss spectroscopy (EELS, Sup. Note 2) and 24.5 and 30% by Rutherford backscattering spectrometry. Notably, the studied concentrations are well beyond the formation threshold of the ordered Fe$_{1-x}$Ga$_x$ intermetallic phase at ~19% Ga, yet our films remain in the A2 phase, showing that epitaxial stabilization allows us to extend the range of the A2 phase[21].

In contrast to previous reports with Fe$_{1-x}$Ga$_x$ thin films[29], when we attempt to anneal our epitaxial films with > 19% Ga, to drive the formation of the thermodynamic phase, we observe degradation of the heterostructure before any structural change in the Fe$_{1-x}$Ga$_x$. Annealing the sample in a vacuum, we see no change in the magnetic properties of the film, which should telegraph any structural change, up to ~ 650 °C. At this temperature, the surface of the PMN-PT begins to degrade, destroying the functionality of the heterostructure.

**Magnetoelectric switching**. To study how the magnetostriction of the Fe$_{1-x}$Ga$_x$ films is influenced by the phase and Ga concentration, we utilize magnetotransport measurements to extract the values from magnetoelectric switching. The films are lithographically patterned into $10 \mu m \times 50 \mu m$ devices, as shown in Fig. 2a, oriented along the $[100]_s$/$[110]_{FeGa}$ direction, the experimentally determined magnetic easy axis of the Fe$_{1-x}$Ga$_x$ (Sup. Fig. 4). Electric fields are applied across the entire substrate thickness using the device as the ground and a back contact for the hot lead. Anisotropic magnetoresistance (AMR) measurements are carried out as a function of angle and magnetic field to determine the direction of the magnetization and magnetic anisotropy. Under an electric field of ±4 kV cm$^{-1}$, low magnetic field (50 Oe) AMR scans show a 90° phase shift of the sinusoidal resistance (Fig. 2b), revealing a 90° separation of magnetization

directions for the two applied electric fields. When saturated at + (−)4 kV cm$^{-1}$, the magnetization lies approximately + (−)45° from the current direction, meaning the magnetization is pulled along the $[110]_s$ ($[1\bar{1}0]_s$) direction. This is the hard axis of the as-grown Fe$_{1-x}$Ga$_x$ layers, indicating that the magnetization direction is dominated by an external voltage-controlled anisotropy. Figure 2c shows the non-volatile electric-field induced 90° magnetization switching, where the direction of the magnetization is relative to the x-direction. From this measurement we infer that the strain from the substrate is non-volatile, oriented along $[110]_s$ and $[1\bar{1}0]$ directions, and depending on the applied voltage, is strong enough to overcome the intrinsic anisotropy barrier of the Fe$_{1-x}$Ga$_x$.

The magnetization direction versus electric-field loop can be used to quantify an effective converse magnetoelectric coefficient, $|\alpha_{eff}|$, of our epitaxial composite multiferroic. We define $|\alpha_{eff}|$ from the expression for the magnetoelectric coefficient, $\alpha = \mu_0 \frac{dM}{dE}$, demonstrative of a magnetization emerging from an applied electric field, to also include the vector rotation of magnetization in the frame of reference of the device, $M = \mathbf{M} \cdot \mathbf{I} = M_S \cos(\phi)$, where $\mathbf{I}$ is the direction of the current. Additionally, we only report the absolute value of this quantity, as the high-field AMR measurements preclude the determination of the handedness of magnetization rotation and the sign of the magnetoelectric coefficient. This definition then allows for $|\alpha_{eff}| = \mu_0 M_S \left| \frac{\partial \cos \phi}{\partial E} \right|$, where $\mu_0$, $M_S$, and $E$ are the vacuum magnetic permeability, the saturation magnetization, and the applied electric field, respectively. Applying this to our magnetoelectric hysteresis loops, the converse magnetoelectric coefficient can reach the giant value of $2.0 \times 10^{-5}$ s m$^{-1}$ in our films with the highest Ga concentration (Fig. 2d).

**Analytical model**. In the past, magnetostriction has been measured by laser interferometry under a magnetic field[30], but the thickness and chemistry of the substrate in our samples essentially preclude this technique. Additionally, clamping of the film to the substrate prevents the use of a strain gauge under a magnetic field, as the substrate should quench any response. The magnetostriction of thin films is typically extracted indirectly from their magnetoelastic coupling coefficients[26] because a direct measure of the static magnetostrictive coefficient is difficult in a thin film geometry due to substrate interaction. Coupling

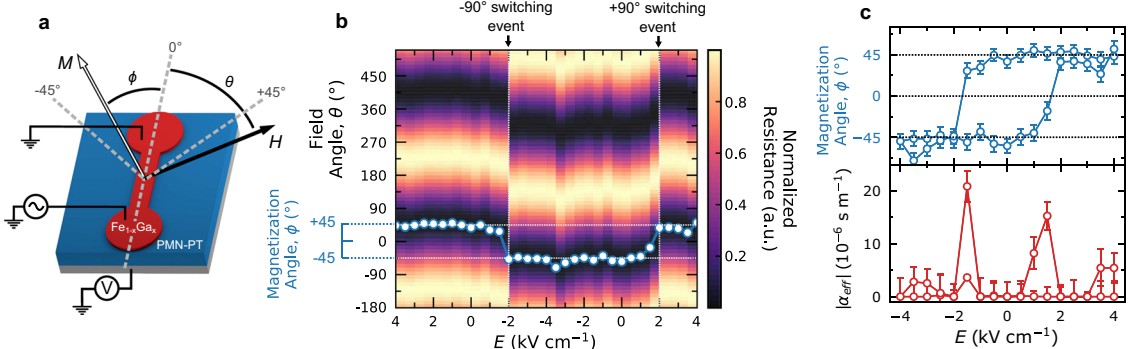

**Fig. 2 Magnetoelectric switching. a** Schematic of the Fe$_{1-x}$Ga$_x$/PMN-PT device. Voltage is applied across the substrate, using the device as a top ground, and resistance is measured along the bar as a function of magnetic field strength and direction ($\phi$). **b** Colormap of low-field (50 Oe) AMR curves fit to $\cos(2\theta)$, showing the normalized resistance as a function of magnetic field direction ($\theta$) and applied electric field. The overlain points correspond to the calculated phase shift from the data, which is the direction of magnetization, $\phi$. The two saturated polarization states of the ferroelectric show a 90° phase shift in the curve, demonstrating a 90° switching of magnetization. **c** Hysteresis of the anisotropy axis, with respect to the direction of the device (x), as a function of electric field, and effective converse magnetoelectric coefficient ($|\alpha_{eff}|$), reaching a maximum value of ~2.0 × 10$^{-5}$ s m$^{-1}$ during switching. The error bars represent the ± 5° angular resolution with one standard deviation of the fit to $\cos(2\theta)$. Parts (**b**) and (**c**) show the representative 30% Ga sample with the largest magnetoelectric coefficient. The full data set is shown in Sup. Fig. S3.

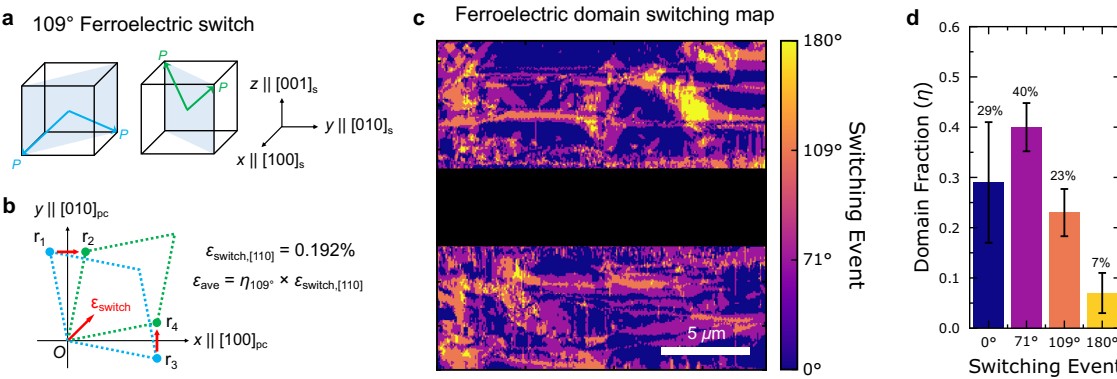

**Fig. 3 Local shear strains arising from 109° polarization switching in PMN-PT. a** Polarizations within the $(110)_s$ (substrate) plane, blue, and the $(\bar{1}10)_s$ plane, green, are associated with shear distortion in the $(001)_{pc}$ plane. **b** indicated by the blue and green dashed frame. The corresponding shear strain arising from a 109° polarization switching can be calculated based on the coordinates of points $r_i$ ($i$=1,2,3,4), where the translation from $r_1$ to $r_2$ results in a 0.192% shear strain per unit cell. This is then scaled by the fraction of ferroelectric domains that undergo a 109° switch ($\eta_{109°}$) to calculate the total strain seen by the device. **c** PFM switching map that allows us to experimentally determine $\eta_{109°}$. This map is made by overlaying PFM micrographs before switching (+4 kV cm⁻¹) and after switching (−4 kV cm⁻¹) and calculating the 3D switching angle per pixel. The directions of the ferroelectric vectors were determined by combining in-plane and out-of-plane piezoresponse patterns before and after rotating the sample by 90° to allow for the determination of in-plane directionality. The full data set is shown in Sup. Fig. S5. **d** Histogram of the switching events from 14 composite images, with standard deviations shown as error bars. The analysis indicates that 23% ± 4% of the domains undergo 109° switchings.

coefficients are normally determined by analyzing the strain-induced change in magnetic anisotropy (or easy-axis reorientation) using analytical models[27,28]. Our approach of determining magnetostriction is similar to these well-established methods[26–28], except that strain is applied dynamically via the piezoelectric layer, which is more convenient than previous experiments that require growing films on different substrates or varying the film thickness to obtain different residual strains. We extract $\lambda_{100}$ from the magnetization versus electric field loop in Fig. 2c, using the energy required to switch the magnetization between the easy and hard axes and the applied strain value.

Associated with an in-plane coherent magnetization switching in the crystallographic reference frame of a (001) $Fe_{1-x}Ga_x$ film, the change in the total magnetic free energy density $\triangle F_{tot}$ can be approximated[31] as

$$\triangle F_{tot} = K_1 m_1^2 m_2^2 + \frac{1}{2}\mu_0 M_s^2\left(N_{11}m_1^2 + N_{22}m_2^2\right) \\ + B_1\left(m_1^2\varepsilon_{11} + m_2^2\varepsilon_{22}\right) + B_2 m_1 m_2 \varepsilon_{12},\quad (1)$$

where $K_1 = −5255, −7434, −6717$ J m⁻³ for the 21.5%, 24.5%, and 30% Ga samples, respectively, is the magnetocrystalline anisotropy constant (extracted from the experimentally measured magnetic hysteresis loops shown in Sup. Fig. S4); $\mu_0$ is the vacuum permeability, and $M_s$ is the saturation magnetization. For our $Fe_{1-x}Ga_x$ film (10 μm × 50 μm × 15 nm in dimension), whose in-plane dimension is much larger than its thickness, the demagnetization energy difference between the short and long axes is calculated from AMR of the virgin sample to be ~0.21 kJ m⁻³. This is ~20× smaller than $K_1$, thus we assume the demagnetization tensor components $N_{11} \cong N_{22} \approx 0$ and the second term in Eq. (1) can be omitted. $B_1 = −1.5\lambda_{100}\left(c_{11} − c_{12}\right)$ and $B_2 = −3\lambda_{111}c_{44}$ are magnetoelastic coupling coefficients (where $\lambda_{100}$ and $\lambda_{111}$ are magnetostrictive coefficients; $c_{11}$, $c_{12}$, and, $c_{44}$ are elastic stiffness coefficients), $m_i = M_i/M_s$ ($i = 1, 2$) are direction cosines of the magnetization vector, and $\varepsilon_{11}$, $\varepsilon_{22}$, $\varepsilon_{12}$ are the average normal and shear strains in the (001) $Fe_{1-x}Ga_x$. Using the PMN-PT substrate as the reference system and assuming complete strain transfer across the coherent PMN-PT/$Fe_{1-x}Ga_x$ interface (see Fig. 1b) without loss, one has $\varepsilon_{11} = \varepsilon_{[1\bar{1}0]_s}$ and $\varepsilon_{22} = \varepsilon_{[110]_s}$.

Writing $\triangle F_{tot}$ as a function of the azimuth angle $\beta$ (via $m_1 = \cos\beta$, $m_2 = \sin\beta$) and minimizing $\triangle F_{tot}(\beta)$ with respect to $\beta$, an analytical formula can be derived for the orientation of the in-plane magnetization vector at equilibrium, denoted as $\beta_{eq}$, as a function of the in-plane strains $\varepsilon_{ij}$ ($i,j = 1,2$). Due to the 45° lattice misalignment between the (100) planes of the $Fe_{1-x}Ga_x$ film and PMN-PT substrate, $\beta = \phi + 45°$ (see definition of $\phi$ in Fig. 1c). Together,

$$\phi_{eq} = \beta_{eq} − 45° = \frac{\cos^{-1}\left(\frac{-B_1\left(\varepsilon_{[110]_s} − \varepsilon_{[1\bar{1}0]_s}\right)}{K_1}\right)}{2} − 45°. \quad (2)$$

Because the experimental realization of the bound values of ±45° has been observed at $E = ±4$ kV cm⁻¹ (Fig. 2c), it is anticipated that the $Fe_{1-x}Ga_x$ film experiences a minimum strain anisotropy of $\varepsilon_{[110]_s} − \varepsilon_{[1\bar{1}0]_s} = −K_1/B_1$ at $E = −4$ kV cm⁻¹ and vice versa. Thus, the magnetostriction coefficient $\lambda_{100}$ can be extracted with knowledge of the strain, $\varepsilon_{[110]_s} − \varepsilon_{[1\bar{1}0]_s}$ at $E = ±4$ kV cm⁻¹ and the pre-factor $\left(c_{11} − c_{12}\right)/2$.

Mechanistically, it has been previously observed that a hysteretic shear strain can be created in $(001)_{pc}$-oriented PMN-PT crystals when the local polarization, **P**, switches by 109° (Fig. 3a) from down ($E = −4$ kV cm⁻¹) to up ($E = 4$ kV cm⁻¹)[32–34]. This distortion corresponding to the downward and upward polarization state is illustrated by the projection of the PMN-PT unit cell onto the $xy$-plane of the substrate (Fig. 3b). Using the lattice parameters and distortion angle of rhombohedral PMN-PT[35], this shear strain is calculated to be 0.192% (see details in Sup. Note 3), which corresponds to biaxial normal strains in the (001) plane of $Fe_{1-x}Ga_x$ along the $[110]_s$ and $[1\bar{1}0]_s$ direction, as indicated in Fig. 3b ($\varepsilon_{switch} = 0.192\%$). Note that this $\varepsilon_{switch}$ only describes the local deformation from the 109° switching of one ferroelectric domain of PMN-PT[36]. The average strain state seen by the $Fe_{1-x}Ga_x$ film is then given by $\varepsilon_{ave} = \varepsilon_{[110]_s} − \varepsilon_{[1\bar{1}0]_s} = \eta_{109}\varepsilon_{switch}$, where $\eta_{109°}$ is the fraction of ferroelectric domains in PMN-PT that undergo the 109° switching (Fig. 3c). Here, we experimentally measure this fraction of 109° switching by comparing composite piezoelectric force microscopy (PFM) micrographs at fields both before and after the magnetoelectric switching event, an example of which is shown in Fig. 3d (full data set shown in Sup. Fig. 5). From the repetition of

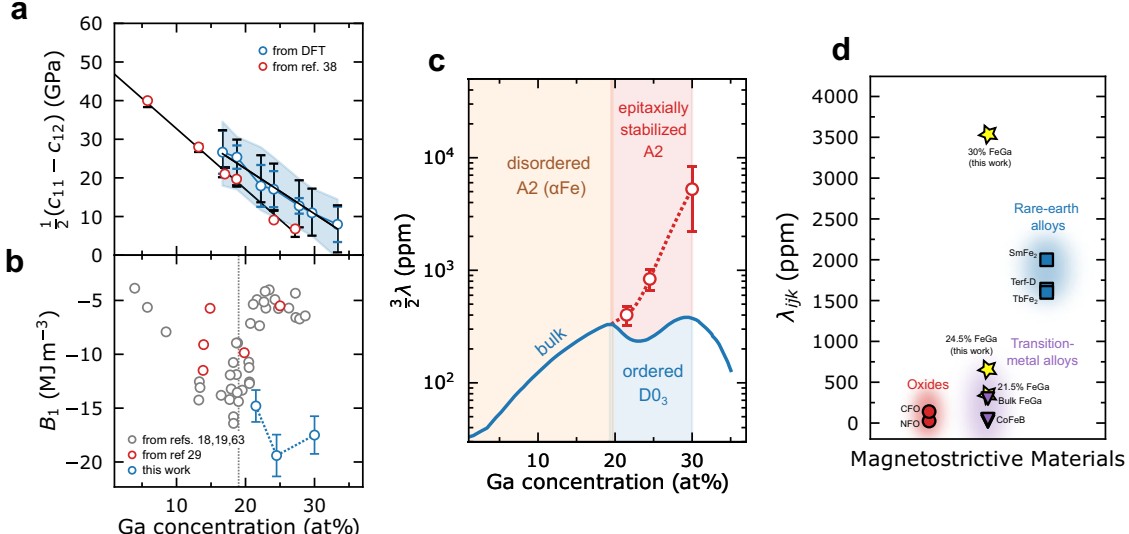

**Fig. 4 Enhanced magnetostriction coefficient through epitaxial stabilization. a** Plot of mechanical coefficient $\frac{1}{2}\left(c_{11} - c_{12}\right)$ extracted from literature (red) and simulated here with DFT (blue). Both data sets follow approximately the same trend and show no deviation from linear behavior following the ~19% phase limit. The blue error bars correspond to the error of the calculation, the black error bars are one standard error of the linear fit, and the shaded area is the sum of the errors fixed about the trendline. Literature values are from ref. [39]. **b** Plot of magnetoelastic coefficients ($B_1$) taken from the previous thin film (red) and bulk (gray) works compared to our measured values. We note that in previous work, there is a sharp decline in $B_1$ following the phase change at ~19% Ga (dotted line), which we do not observe. Bulk (gray) values are from refs. [18,19,63] and film (red) values are from ref. [30]. **c** The extracted magnetostriction values as a function of Ga concentration with our values (red, open circles) compared to the measured bulk coefficients (blue) from ref. [19]. The values from this work are plotted as $\frac{3}{2}\lambda_{100}$ to facilitate comparison with the bulk, polycrystalline values. Above 19% Ga, we do not observe a decrease in the magnetostriction associated with the formation of the ordered D0₃ phase and we extend the regime of the disordered A2 phase via epitaxial stabilization[21]. As the concentration approaches the second phase change at ~30% Ga, the shear modulus $c_{11} - c_{12}$ approaches 0, leading to extremely large values of the magnetostriction. Calculation of the error bars, $\sigma_\lambda$, is detailed in Sup. Note 4. **d** Comparison of the magnetostriction coefficients from this work to other magnetostrictive materials. The largest magnetostrictive tensor component $\lambda_{ijk}$ of each respective material is plotted here for ease of comparison. Comparative data in (**b**) from refs. [54-62].

this measurement (Fig. 3e), we estimate that $\eta_{109} \cong 23\% \pm 4\%$, consistent with published values in $(001)_{pc}$ PMN-PT crystals[32-34] determined through in situ reciprocal space mapping.

In bulk $Fe_{1-x}Ga_x$, the pre-factor $(c_{11} - c_{12})/2$ can range from 28 GPa to 7 GPa due to the variation of the Ga composition[19,37] (from 17% to 27.2%). $(c_{11} - c_{12})/2$ values from DFT simulation of disordered $Fe_{1-x}Ga_x$ thin films agree with published experimental bulk values from Clark et al.[19] to within the error bar of the simulation (Fig. 4a), justifying the use of bulk compliance elements in our analysis here. Additionally, these bulk values appear to be constant between phases, as $(c_{11} - c_{12})/2$ follows a linear trend across the 19% Ga threshold, and is free of an anomaly that would correspond to the phase change in bulk. This indicates that the nature of the magnetostrictive transition is electronic and manifests as a deviation of $B_1$ from the bulk value. Using these values for $\varepsilon_{22} - \varepsilon_{11}$ and $(c_{11} - c_{12})/2$, we calculate a $\lambda_{100}$ for the system between ranging between 300 ppm (21.5% Ga) and 3500 ppm (30% Ga sample), significantly higher than reported bulk values[19] ($\lambda_{100} \cong 200$ ppm). Comparatively, previously published thin-film results[30,38] indicate a bulk-like behavior in $Fe_{1-x}Ga_x$ grown epitaxially on GaAs substrates. Compositions studied near the phase boundary at 19% Ga show relative agreement between measured magnetoelastic ($B_1$) coefficients. Samples studied here, however, deviate significantly from bulk behavior after 19% and the large discrepancy appears to be due to the $B_1$ coefficient which, in these previous reports, shows a sharp drop after the 19% Ga threshold, commensurate with the bulk decrease in $\lambda$ (Fig. 4b). We do not observe this phenomenon in our samples, leading to a large enhancement in magnetostriction considering bulk mechanical data, which

follows a linear trend that is irrespective of phase[39]. This enhancement of the magnetoelasticity, and thus magnetostriction, demonstrates our ability to stabilize the A2 phase far beyond the bulk threshold in our system[20]. Furthermore, these values are plotted against the $\lambda_{100}$ values for bulk $Fe_{1-x}Ga_x$ and we note that our peak value of magnetostriction coincides with the lattice softening at ~30% Ga (Fig. 4c). This implies that we are not only able to prevent the formation of the parasitic intermetallic phase[16], but through epitaxial engineering, can leverage the inherent phase space to reach record values. Comparing these numbers to existing magnetostrictive materials in Fig. 4d, we see that our peak values are ~2× higher than top-performing, rare-earth-based magnetostrictive alloys. We note that this enormous enhancement of the magnetostriction is largely a function of the mechanical coefficients, as the magnetoelastic energy, $B_1$, remains largely invariant.

To gauge and benchmark technological performance, we can consider the energy dissipation of the switching as the scale of the device is decreased. The area-normalized energy dissipation per switch determined by integrating half of the ferroelectric polarization loop (Fig. 3d), corresponding to a single switch, is 2.9 μJ cm⁻². Scaling the ferroelectric layer thickness to 100 nm, the energy dissipation per switch would drop to 5.9 μJ cm⁻² (using the coercive field of 30 kV cm⁻¹ reported in ref. [40]), giving this system the best scaling projection for normalized energy dissipation per switch of any composite multiferroic[8]. At this thickness, substrate clamping is well known to suppress piezo-electric response, however, clamping to the substrate can be minimized through lithography such as patterning[41] or exfoliation[42]. The latter has also been shown to reduce the coercive voltage, relative to clamped films, of the ferroelectric[40].

Combining this with an idealized magnetic device of $45 \times 45$ nm$^2$, the smallest magnet size to preserve thermal stability[5] ($>42\ k_\mathrm{B}T$) based on experimental anisotropy, this ideal bit would have a switching energy of ~80 aJ, making our devices competitive with other state-of-the-art beyond-CMOS technologies[5]. As this analysis only considers the energy required to switch the ferroelectric, we have also used micromagnetic simulation to predict the energy of magnetization reorientation. For an idealized 45 nm device, the calculated magnetic energy dissipation is <1 aJ per 90° magnetic switch (Sup. Fig. S7).

In conclusion, our results demonstrate that significantly higher performance magnetostrictors can be achieved in earth-abundant, rare-earth-free materials by using epitaxy to extend the stability of the desired phase[21]. The improved magnetostriction translates into unparalleled performance for a composite multiferroic in non-volatile computing and memory applications.

## Methods

**Sample preparation and growth.** The $Fe_{1-x}Ga_x$ thin films presented were grown by molecular-beam epitaxy in a Veeco GEN10 system on (001)-oriented (0.70) $PbMg_{1/3}Nb_{2/3}O_3 - (0.30)\ PbTiO_3$, PMN-PT, 1 mm substrates held at a temperature of 160 °C. The substrates were preheated to ~ 375 °C for cleaning. Iron and gallium fluxes were determined by a quartz crystal microbalance. Typical fluxes from the elemental sources were ~1–2.7 × 10$^{13}$ Fe atoms cm$^{-2}$ s$^{-1}$ and 0.2−0.6 × 10$^{13}$ Ga atoms cm$^{-2}$ s$^{-1}$. The $Fe_{1-x}Ga_x$ films were deposited at a substrate temperature of 160 °C after a ~3 unit cell thick layer of Fe was deposited at the same temperature to seed the growth. The orientation and phase purity of the deposition of the epitaxial $Fe_{1-x}Ga_x$ was monitored with in situ refractive high-energy electron diffraction (RHEED). In situ RHEED images along the [110]$_s$ azimuth of PMN-PT before $Fe_{1-x}Ga_x$ deposition and along the [100] azimuth of $Fe_{1-x}Ga_x$ after deposition are shown in Sup. Fig. S1. The diffraction streaks reveal that the films are smooth and have four-fold in-plane symmetry. RHEED also reveals the epitaxial orientation relationship of [110]$_s$ PMN-PT//[100] $Fe_{1-x}Ga_x$, thus, the cubic $Fe_{1-x}Ga_x$ grows at an in-plane 45° rotation with respect to the PMN-PT cubic unit cell. Following the deposition, the film is cooled to below 60 °C where then the films were capped with ~5 nm of Ti.

**Rutherford backscattering spectroscopy.** RBS with 1.4 MeV He 4+ ions was used to assess the stoichiometry of the films. The results were analyzed using the software program RUMP[43].

**Electron microscopy.** High-angle annular dark-field scanning transmission electron microscopy (HAADF-STEM), electron energy loss spectroscopy (EELS) and energy-dispersive X-ray spectroscopy (EDS) data were collected using JEOL3100R05 aberration corrected STEM equipped with cold field emission gun, gatan quantum imaging filter, and JEOL silicon drift detector. STEM was operated at 300 keV with 22 mrad convergence angle. EELS and EDS spectra were acquired with energy 0.25 eV and 10 eV dispersion respectively, with EELS collection angle of 34 mrad. A cross-sectional TEM sample was prepared by focused ion beam (FIB) lift out on FEI Helios 650 Nanolab DualBeam.

Relative Ga/Fe concentration was quantified by taking the ratio between integrated core-loss intensity and Hartree-Slater calculated inelastic scattering cross section of Fe/Ga L - edge. Single scattering core-loss EELS spectra were extracted by Fourier-ratio deconvolution. (See Sup. Fig. S2 and Note 1) EDS quantification of the relative concentration used the Cliff-Lorimer ratio method on Fe and Ga $K_\alpha$ peaks. Convergence and collection angles were calibrated before quantification.

**Lithography.** Thin film samples were patterned using photolithography at the University of Michigan Lurie Nanofabrication Facility. Mask designs were written on 4 fused silica mask plates using a Heidelberg DWL-2000 mask writer with a 4 mm write head. The photolithography for the ion milling and deposition patterns was done using a conventional SPR 220 based process. Samples were then ion milled using a conventional Ar ion mill and 20 nm Ti/ 100 nm Pt contacts were deposited by sputtering after a second photolithography step. Samples were bonded to a custom-built PCB for transport measurements using a gold wire.

**Transport.** AMR was measured using a custom-built electronic transport rotator with $\chi$ and $\phi$ rotation capabilities and a Lakeshore EM4-HVA electromagnet. Low-magnetic-field scans were measured by saturating the magnetization along the easy axis in a 2 kOe field and then rotating the sample in-plane under a constant 50 Oe magnetic field while measuring longitudinal resistance using a Keithley 2420 source meter. High-magnetic-field scans were taken by fixing the sample in-plane along a particular crystallographic direction and sweeping the magnetic field from −2 to 2 kOe while measuring longitudinal resistance. The saturation resistances were then normalized by rotating the sample in-plane in a constant 5 kOe magnetic field

and fitting to $R = R_{mean} + \Delta R\cos\phi$ to determine $R_\parallel$ and $R_\perp$. Successive voltages (400 V to −400 V) were applied to the samples using a Keithley 2410 sourcemeter. Samples were cycled several times in voltage before measurement to eliminate potential artifacts from ferroelectric domain pinning. Resistances for in-text Figures were normalized by scaling from 0 to 1.

**Effective magnetoelectric coefficient determination.** The effective converse magnetoelectric coefficient, $|\alpha_{eff}|$, is given by

$$|\alpha_{eff}| = \mu_0 \left| \frac{dM}{dE} \right| = \mu_0 M_S \left| \frac{d\cos\phi}{dE} \right|,$$

where $\mu_0$ is the permeability of free space, $M_s$ (980 emu cm$^{-3}$ by VSM) is the saturation magnetization of the $Fe_{1-x}Ga_x$, $E$ is the applied electric field, and $\phi$ is the angle the magnetization makes with the direction of the applied current. $|\alpha_{eff}|$ was calculated by taking the numerical derivative of the magnetoelectric hysteresis loops shown in Fig. 2c and Sup. Fig. S3.

**Ferroelectric loops.** Polarization versus applied electric field loops was taken on a Radiant Precision MF2 ferroelectric tester using a frequency of 1 kHz. Loops were reconstructed from PUND measurements to remove electronic artifacts due to leakage. Integration of this loop multiplied by thickness was used to determine the energy per switch per unit area of our device.

**Determination of magnetic anisotropy.** Magnetometry of samples was taken along different crystallographic directions using a Lakeshore vibrating sample magnetometer (VSM) with in-plane rotation. Integration of moment versus field curves with $A_{ijk} = \int_0^{H(M_S)} M\ dH$ yields the energies between the [100] and [110] crystallographic directions, which can be related to the anisotropy energy $K_1$ through $A_{110} - A_{100} = K_1/4$ (See Sup. Fig. S4). Differences in thin film and bulk magnetization values can be due to epitaxial strain, as it has been observed in bulk that residual strain can affect the value of the magnetization[44].

**Piezoresponse force microscopy.** PFM was done on an NT-MDT Ntegra Prima microscope using Pt coated tips. A ~10 V, 1074 Hz AC voltage was applied across the samples and both the in-plane and out-of-plane deflection of the tip was measured to construct domain patterns. To construct the composite switching maps, the sample is first poled at +400 V and PFM is done. The combination of in-plane and out-of-plane PFM responses can be used to map the direction of polarization in 3D space. The sample is then poled at −400 V and the same PFM mapping is done. The two images, +400 V and −400 V are then compared pixel-by-pixel to create a map of the switching events where the color corresponds to the angle between the polarization vectors of before and after images. For example, if a pixel maps to a (+x,+y,+z) polarization after +400 V, then a (+x,+y,-z) polarization after −400 V, that corresponds to a 71° out-of-plane switching event.

**Determination of idealized energetics.** The ferroelectric energy per switch per area ($\Delta E_{FE} = 2.9$ mJ cm$^{-2}$) was calculated by taking 1/2 of the integral of the ferroelectric hysteresis loop shown in Fig. 1d, corresponding to approximately $2 * P_r * V_c$. This value is reduced to an idealized thickness (100 nm) and scaled by the increased coercive field reported in reference[40], in order to translate to more applicable device geometries, to 5.87 $\mu$J cm$^{-2}$. The magnetic anisotropy energy, $K_1$, between the [100]-easy and [110]-hard magnetic axes was measured to be −5255, −7434, −6717 J m$^{-3}$ for the 21.5%, 24.5%, and 30% Ga samples, respectively (Sup. Fig. S4). In our 15 nm-thick magnet, this energy can be evaluated as a function of length scale ($l$) for simple circular ($A = \pi\left(\frac{l}{2}\right)^2$) devices. When this energy reaches 42 $k_B T$, generally accepted as the minimum energy required for directional coherence, we reach a minimum device size of ~45 nm. Using this to calculate the area for the ferroelectric energy dissipation at scale, we find an energy consumption of ~80 aJ per switch.

**Density functional theory calculations.** We performed DFT calculations based on the projector augmented wave (PAW) method[45,46], as implemented in the Vienna Ab initio Simulation Package (VASP)[47-50]. Random alloys were modeled using special quasi-random structures (SQSs) generated with the Alloy Theoretic Automated Toolkit[51] as $2 \times 2 \times 2$ and $3 \times 3 \times 3$ supercells. Atoms were arranged to approximate the pair-correlation functions of random alloys up to 5 Å. 14 and 13 valence electrons were included for Fe and Ga, respectively, with a 500 eV plane-wave cutoff and $5 \times 5 \times 5$ and $3 \times 3 \times 3$ Monkhorst-Pack[52] Brillouin-zone-sampling grid for the 16 atoms and 54 atoms respectively. Forces on atoms were relaxed to within 5 meV Å$^{-1}$ and electronic convergence set to within 10$^{-6}$ eV. Calculations were performed using collinear magnetization. Elastic properties were calculated using isotropic volume expansion to obtain the bulk modulus, epitaxial strain to obtain $c_{11}$ and $c_{12}$, and volume-conserving shear strain to obtain $c_{44}$[53]. Room-temperature values of $c_{11}$ and $c_{12}$ were approximated from the temperature dependence of experimental values in[39].

**Micromagnetic simulations**. Please see Sup. Note S3 for details.

## Data availability

All data needed to evaluate the conclusions in the paper are present in the paper and/or the Supplementary Materials. Additional data related to this paper may be requested from the authors.

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

## Acknowledgements

This work was funded in part by IMRA America. This work was performed in part at the Cornell NanoScale Science & Technology Facility and at the University of Michigan Lurie Nanofabrication Facility, members of the National Nanotechnology Coordinated Infrastructure (NNCI), which is supported by the National Science Foundation (NSF) through Grant NNCI-1542081. This work was partially supported by the NSF (Nanosystems Engineering Research Center for Translational Applications of Nanoscale Multiferroic Systems) under Grant Number EEC-1160504, made use of the Cornell Center for Materials Research Shared Facilities supported through the NSF MRSEC program (DMR-1719875), and Michigan Center for Materials Characterization. Materials synthesis was performed in a facility supported by the NSF (Platform for the Accelerated Realization, Analysis, and Discovery of Interface Materials (PARADIM)) under Cooperative Agreement No. DMR-1539918. This work was supported in part by the Semiconductor Research Corporation (SRC) as the NEW-LIMITS Center and NIST through award number 70NANB17H041. J.-M.H. acknowledges support from NSF grant CBET-2006028. This work used Bridges at the Pittsburgh Supercomputing Center through allocation TG-DMR180076, which is part of the Extreme Science and Engineering Discovery Environment (XSEDE) and supported by NSF grant ACI-1548562. R.H. acknowledges funding from National Science Foundation grant DMR-1807984. M.M.T. and S.A.B. acknowledge support from SRC contract # 2018-LM-2830.

## Author contributions

P.B.M. performed the magnetic and dielectric measurements. P.B.M., S.N., C.J.Z., M.M.T., and B.P. fabricated the magnetoelectric test structures. P.B.M. and M.E.N. performed magnetotransport measurements and analysis. S.H.S. and A.H.H. performed the electron microscopy measurements and, along with R.H., performed the analysis of electron microscopy measurements. L.D.W. performed DFT calculations. S.Z. and J.-M.H. performed the analytical calculations and micromagnetic simulations. R.A.S., N.M.D., and Z.W. grew the epitaxial films. P.B.M., S.M., D.N., I.Y., J.B., L.Q.C., D.G.S., R.R., and J.T.H. directed elements of the study and analyzed results. P.B.M., S.H.S., L.D.W., S.Z., R.R., J.-M.H., R.H., E.K., D.G.S., and J.T.H. co-wrote the manuscript. All authors reviewed the manuscript.

## Competing interests

The authors declare no competing interests.
