## [Peer Review File · Nature Communications]

Reviewers' Comments:

Reviewer #1:

Remarks to the Author:

The authors in the presented paper discovered that the epitaxially prepared Fe_{1-x}Ga_x alloys can be increase the magnetostriction not only upto 18% of x but upto 30%. In the epitaxially prepared samples they obtained very large magnetostriction in the epitaxially stabilized A2 phase. This is very new results and the paper can be published as is.

Reviewer #2:

Remarks to the Author:

This work claimed harness epitaxy to extend the stability of the BCC Fe_{1-x}Ga_x alloy to gallium compositions as high as x=30% and in so doing dramatically boost the magnetostriction by as much as 20x relative to the bulk and 2x larger than canonical rare-earth based magnetostrictors. A Fe_{1-x}Ga_x - [Pb(Mg_{1/3}Nb_{2/3})O₃]_{0.7}-[PbTiO₃]_{0.3} (PMNPT) composite magnetoelectric shows robust 90° electrical switching of magnetic anisotropy and a converse magnetoelectric coefficient of 2.0×10⁻⁵ s m⁻¹. The results obtained are very interesting and attracting. I have the following question: In this work, the magnetostriction of thin films is typically extracted indirectly from their magnetoelastic coupling coefficients, with several assumptions and with many parameters for calculations. Since the very high magnetostriction of BCC Fe_{1-x}Ga_x thin films is the most important result of this work, it would be much convince to measure the magnetostriction of these thin films using direct methods.

Reviewer #3:

Remarks to the Author:

In this article, the authors claim to have grown FeGa thin films presenting an A2 (or rather, a D03 free) phase such that magnetostriction is strongly enhanced. The values of magnetostriction reported in the article are very high for a rare -earth-free compound. If confirmed, this would be a new and very interesting claim that could warrant a publication on Nat Comm. Indeed, there is a lot of interest for rare earth free hybrid piezoelectric/magnetostrictive devices for energy harvesting and sensors applications.

By the way, I'm not convinced by the main claims of the authors and some precisions warrant to be made.

First of all, it is well known that epitaxially FeGa thin film present good magnetostrictive properties with values close to the bulk's one. See for example : Parkes et al. Scientific Reports 3, Article number: 2220 (2013) and Barturen et al. PRB 99, 134432 (2019).

So the main questions are: did the authors really measure a huge increase of the magnetostrictive properties, i.e. well beyond the bulk values, and is this due to the A2 structure that they stabilized?

Concerning the first question, my main concern is the following: the authors do not report the error bars in Fig.4 a. and this is a pity. If I well understood, λ_{100} is obtained by the ratio between the magnetic anisotropy constant K and the average strain dot the elastic constant difference.

I estimate the relative error on K to be around 5-10% (mainly due to the thickness evaluation of the sample), the relative error on the average strain around 50% (the 20% of fraction of ferroelectric is approximative and epsilon is not measured on their samples) and a big error (up to 100%) on the elastic constants difference, $c_{11}-c_{12}$. Indeed, the values of the elastic constants are given without errors bar by adopting the measurements by Clark, corroborated by ab initio calculations (for which the error bars are shown). Now, $c_{11}-c_{12}$ is very low for XGa around 30%...the relative error bar is very high. Moreover, ab initio calculations describe the $c_{11}-c_{12}$

trend but fail in describing the C11 experimental points (the authors do not mention this point). Consequently, the error bar on λ_{100} is around 170%! Only a precise experimental measurement of C11-C12 could clearly put into evidence such high values of λ_{100} . I'd appreciate that the authors evaluate the error bars.

Concerning the claim about the pure A2 phase, my main concern is the following. In Fig. S1 the authors show a diffraction diagram (much better than TEM figures that present a local and partial view). The [001] diffraction peak is not observed. This indicates that the B2 and the D03 phases, if they exist, are not well ordered. It's very likely that nanoclusters with Ga pairs are present in the samples, as discussed in ref.17.

My main comments are :

- 1) The authors should give more details about the Ga concentration of data reported in Fig S.1
- 2) Did the authors succeed in obtaining samples with B02 or D03 structures (ex. by annealing). If yes, did they try to evaluate the magnetostriction properties and the magnetoelectric coefficient? this would be a clear evidence of the importance of the atomic structure to enhance those properties.
- 3) Did the authors give a look to the [113] diffraction peaks that are typical of the D03 structure?

Minor concerns:

- 1) which is the thickness of the substrates?
- 2) Authors say that the D03 structure is bcc. I'd rather say that it is bcc-like.
- 3) Did the authors try to follow magnetization by Kerr when the voltage is applied on the magnetoelectric device? The authors suppose that the system is single domain when they evaluate α_{eff} , don't they?
- 4) What is the estimated error bar on α_{eff} ?
- 5) I did not understand how Figure 3 d is obtained. It's not clear for me, sorry.

Reviewer #1 (Remarks to the Author):

The authors in the presented paper discovered that the epitaxial prepared Fe_{1-x}Ga_x alloys can be increase the magnetostriction not only upto 18% of x but upto 30%. In the epitaxial prepared samples they obtained very large magnetostriction in the epitaxially stabilized A2 phase. This is very new results and the paper can be published as is.

We thank the reviewer for their positive evaluation of our manuscript.

Reviewer #2 (Remarks to the Author):

This work claimed harness epitaxy to extend the stability of the BCC Fe_{1-x}Ga_x alloy to gallium compositions as high as x=30% and in so doing dramatically boost the magnetostriction by as much as 20x relative to the bulk and 2x larger than canonical rare-earth based magnetostrictors. A Fe_{1-x}Ga_x - [Pb(Mg_{1/3}Nb_{2/3})O₃]_{0.7}-[PbTiO₃]_{0.3} (PMNPT) composite magnetoelectric shows robust 90° electrical switching of magnetic anisotropy and a converse magnetoelectric coefficient of 2.0×10⁻⁵ s m⁻¹. The results obtained are very interesting and attracting. I have the following question: In this work, the magnetostriction of thin films is typically extracted indirectly from their magnetoelastic coupling coefficients, with several assumptions and with many parameters for calculations. Since the very high magnetostriction of BCC Fe_{1-x}Ga_x thin films is the most important result of this work, it would be much convince to measure the magnetostriction of these thin films using direct methods.

We thank the reviewer for their positive evaluation of our manuscript. From our study, our lowest Ga composition (21.5%) has values of magnetoelasticity (B_1) and magnetostriction (λ_{100}) that are comparable to previous reports of both bulk ² and thin film samples ¹ at similar concentrations (Fig. R1a). The reproducibility is a validation of our approach. However, we agree that a direct measure of magnetostriction would be valuable. Unfortunately, a direct measure of the static magnetoelectric coefficient is very difficult in a thin film geometry due to substrate effects and the axis of measurement. Magnetostriction is typically measured directly by laser interferometry under a magnetic field ¹, but the in-plane magnetization and strain of our samples largely invalidates this technique. Additionally, clamping of the film to the substrate prevents the use of a strain gauge under magnetic field, as the substrate will not respond and should quench any response. Our approach opens new possibilities for assessment of thin film magnetostriction.

Figure R1 | **Plot of magnetoelastic and magnetostrictive coefficients.** **a**, Here, we plot the magnetoelastic coefficient, B_1 , from reference ¹ against data from our study. Bulk values (grey) are from refs. ²⁻⁴ and film values are from ref. ¹. Values near our lowest studied Ga concentration, 21.5%, are comparable, providing a comparative measurement and revealing that our method is reasonable. Points measured in this work (blue) are shown with a 10% relative uncertainty. **b**, Phase diagram of showing magnetostriction coefficient, λ_{100} , as a function of Ga concentration, with bulk values from ref. ² shown in blue and values from this work shown in red.

Reviewer #3 (Remarks to the Author):

In this article, the authors claim to have grown FeGa thin films presenting an A2 (or rather, a D0₃ free) phase such that magnetostriction is strongly enhanced. The values of magnetostriction reported in the article are very high for a rare-earth-free compound. If confirmed, this would be a new and very interesting claim that could warrant a publication on Nat Comm. Indeed, there is a lot of interest for rare-earth-free hybrid piezoelectric/magnetostrictive devices for energy harvesting and sensors applications. By the way, I'm not convinced by the main claims of the authors and some precisions warrant to be made.

First of all, it is well known that epitaxial FeGa thin film present good magnetostrictive properties with values close to the bulk's one. See for example : Parkes et al. Scientific Reports 3, Article number: 2220 (2013) and Barturen et al. PRB 99, 134432 (2019). So the main questions are: did the authors really measure a huge increase of the magnetostrictive properties, i.e. well beyond the bulk values, and is this due to the A2 structure that they stabilized?

We thank the reviewer for their insightful comments. While the papers mentioned here do characterize the magnetostriction of FeGa thin films, their results inform ours but do not supersede them. In the paper by Parkes et al., magnetotransport in the form of the planar Hall effect and a free energy macrospin model is used to extract the magnetoelastic coefficient. The methodology is similar to ours. Their calculated values of B_1 are approximately 25 MJ/m^3 , which is comparable to our values. They, however, do not explore any compositions other than 19.5% Ga. In the paper from Barturen et al., they determine the magnetoelastic coefficients by fabricating a cantilever device out of a semiconducting substrate and actuating it with a magnetic field. This is not directly comparable to our structure because their material above 19% appears to be in the D03 phase. While their values of B_1 are similar to ours ($\sim 16 \text{ MJ/m}^3$) at $\sim 19\%$ Ga, their B_1 values drop sharply as Ga concentration is increased from there and consistent with trends in bulk indicating the formation of the D03 phase. B_1 values from our measurements remain approximately constant for our composition series indicating no emergence of a secondary phase. This is illustrated below in **Figure R2**.

Figure R2 | Plot of magnetoelastic coefficient. Here, we plot the magnetoelastic coefficient, B_1 , from reference ¹ mentioned above against data from our study. Bulk values (grey) are from refs. ²⁻⁴ and film values are from ref. ¹. While peak values are similar to what is observed in our work, ($\sim 16 \text{ MJ/m}^3$) at $\sim 19\%$ Ga, B_1 values from reference ¹ drop sharply as Ga concentration is increased above the phase threshold of $\sim 19\%$, while ours remain around 18 MJ/m^3 . This implies that, while their compositional series follows bulk trends and magnetostriction drops when the D0₃ phase begins to form, our samples remain in the disordered phase and magnetoelastic coupling remains high. This, combined with quickly decreasing mechanical coefficients, enables large values of λ_{100} . Points measured in this work (blue) are shown with a 10% relative uncertainty.

Concerning the first question, my main concern is the following: the authors do not report the error bars in Fig.4 a. and this is a pity. If I well understood, λ_{100} is obtained by the ratio between the magnetic anisotropy constant K and the average strain dot the elastic constant difference. I estimate the relative error on K to be around 5-10% (mainly due to the thickness evaluation of the sample), the relative error on the average strain around 50% (the 20% of fraction of ferroelectric is approximative and epsilon is not measured on their samples) and a big error (up to 100%) on the elastic constants difference, $c_{11}-c_{12}$. Indeed, the values of the elastic constants are given without errors bar by adopting the measurements by Clark, corroborated by ab initio calculations (for which the error bars are shown). Now, $c_{11}-c_{12}$ is very low for XGa around 30%...the relative error bar is very high. Moreover, ab initio calculations describe the $c_{11}-c_{12}$ trend but fail in describing the c_{11} experimental points (the authors do not mention this point). Consequently, the error bar on λ_{100} is around 170%! Only a precise experimental measurement of $c_{11}-c_{12}$ could clearly put into evidence such high values of λ_{100} . I'd appreciate that the authors evaluate the errors bars.

As stated by the reviewer, the primary sources of error in our interpretation are 1) the magnetocrystalline anisotropy, 2) the values for strain, and 3) the stiffness tensor components c_{11} and c_{12} .

Regarding 1: We agree that a 5-10% estimated error would likely be appropriate due to tool calibration/misalignment.

Regarding 2: As we do not have a statistically significant number of scans, a way we can empirically determine the uncertainty associated with the switching fraction is by comparison to bulk values. In refs. ^{5,6}, the authors report 20% 109° switching and ref. ⁷ reports 26% switching. Including our values, the standard deviation $\sigma_\eta = 2.6\%$, which is used to calculate the error bars in λ .

Regarding 3: We use the bulk values of c_{11} and c_{12} because we believe that they present a more accurate estimate of the real stiffness coefficients than can be obtained from DFT calculations. While the DFT reproduces the trend and approximate order of magnitude well, we do not believe the values themselves are accurate due to several challenges. As we consider the A2 phase of FeGa which is a disordered alloy, a very large supercell must be used to accurately predict these values, which past a certain point becomes prohibitively computationally expensive. Additionally, possible atomic arrangements of the species, such as atomic scale Ga-Ga ordering, may have a significant effect on mechanical properties, as suggested by refs. ^{8,9} and our own unpublished results. Since we don't know which of large number of potential supercells can match experiment most accurately, and since there are sizable differences in

values between them, this introduces a source of error that does not exist in the experimental results.

There is precedent in the literature that the large error in the calculation is surmountable with enough computational resources (and thus should not be used to hold back the error analysis of the experimental data). The bulk structures in ref. ⁸ have very large variations in their individual supercell's properties, but their weighted averages at each composition gave clean trends across composition space. Fundamentally, the issues in the values from DFT come from a difficulty in sampling microstructural possibilities in computation, which is not something that exists in the real-world measurement of the property, so the real-world error bars should be used.

As further motivation for comparison to bulk values, it was found in ref. ¹, mentioned above by the reviewer, that thin-film magnetoelastic coefficients approach those of bulk in the sub-19% A2 regime, providing a precedent for their use in our novel phase space.

As no uncertainty values are reported in ref. ², where we obtain the c_{11} , c_{12} values, we use the standard error of the linear trendline in **Sup. Figure S6b** as the error for $\frac{(c_{11}-c_{22})}{2}$.

The uncertainty in λ is defined as

$$\sigma_{\lambda}^2 = \left[\frac{1}{3} \frac{K_1}{\epsilon} \left(\frac{1}{c^2} \right) \right]^2 \sigma_c^2 + \left[\frac{1}{3c\epsilon} \right]^2 \sigma_{K_1}^2 + \left[\frac{1}{3} \frac{K_1}{c} \left(\frac{1}{\epsilon^2} \right) \right]^2 \sigma_{\epsilon}^2,$$

where $\epsilon = \eta \epsilon_{switch}$, and $c = \frac{c_{11}-c_{12}}{2}$, and we can then plot error bars on **Figure 4a/R3c**.

Figure R3 | Detailed calculation of σ_λ . **a**, Density functional theory (DFT) calculations of $\frac{c_{11}-c_{12}}{2}$ (blue), plotted against experimental $\frac{c_{11}-c_{12}}{2}$ values taken from ref. ² (red). The error bars on the blue points correspond to the population standard deviation of the calculations (blue error bars), with the error of a linear fit for the individual components (black error bars) and the combination of the two (shaded area). The error bars on the red points correspond to the standard error of the linear trendline shown in black; these are used in the calculation of λ , for reasons discussed above. **b**, Solutions to the analytical model for λ , for each composition, plotted against switching fraction and corresponding strain, with error bars corresponding to the σ_λ shown above. The values used are $\sigma_c \cong 1$ GPa, $\sigma_{K_1} = 10$ rel%, and $\sigma_\epsilon = 13$ rel%= 2.6%. **c**, The same error bars shown in **b**, extended to the phase diagram shown in **Figure 4a**.

This error analysis is also described in **Sup. Note 4** as shown below.

Supplementary Note 4 | Detailed uncertainty in λ_{100}

The primary sources of error in the calculation of λ_{100} are 1) the magnetocrystalline anisotropy, 2) the values for strain, and 3) the stiffness tensor components c_{11} and c_{12} . For the anisotropy constant K_1 , a 5% relative error is appropriate from tool calibration/misalignment. Regarding the switching fraction, we can empirically determine the uncertainty by comparison to bulk values. In refs. ^{5,6}, the authors report 20% 109° switching and ref. ⁷ reports 26% switching. Including our values, the standard deviation $\sigma_\eta = 2.6\%$, which is used to calculate the error bars in λ .

We use the bulk values of c_{11} and c_{12} because we believe that they present a more accurate estimate of the real stiffness coefficients than can be obtained from DFT calculations. While the DFT reproduces the trend and approximate order of magnitude well, the approximations involved in the calculations introduce systematic errors, motivating us to pull more precise values from literature. As no uncertainty values are reported in ref. ², where we obtain the c_{11} , c_{12} values, we use the standard error of the linear trendline in **Sup. Figure S6b** as the error for $\frac{(c_{11}-c_{22})}{2}$. The uncertainty in λ_{100} can then be defined as

$$\sigma_\lambda^2 = \left[\frac{1}{3} \frac{K_1}{\epsilon} \left(\frac{1}{c^2} \right) \right]^2 \sigma_c^2 + \left[\frac{1}{3c\epsilon} \right]^2 \sigma_{K_1}^2 + \left[\frac{1}{3} \frac{K_1}{c} \left(\frac{1}{\epsilon^2} \right) \right]^2 \sigma_\epsilon^2,$$

where $\epsilon = \eta \epsilon_{switch}$, and $c = \frac{c_{11}-c_{12}}{2}$.

Concerning the claim about the pure A2 phase, my main concern is the following. In Fig. S1 the authors show a diffraction diagram (much better than TEM figures that present a local and partial view). The [001] diffraction peak is not observed. This indicates that the B2 and the D03 phases, if they exist, are not well ordered. It's very likely that nanoclusters with Ga pairs are present in the samples, as discussed in ref.17.

My main comments are:

1) The authors should give more details about the Ga concentration of data reported in Fig S.1

The following has been added to the caption of Sup. Figure S1: “The data shown correspond to the 18.5% Ga film. The other studied compositions are equivalent and as such are omitted.”

2) Did the authors succeed in obtaining samples with BO2 or D03 structures (ex. by annealing). If yes, did they try to evaluate the magnetostriction properties and the magnetoelectric coefficient? this would be a clear evidence of the importance of the atomic structure to enhance those properties.

From our study, our lowest Ga composition (21.5%) has values of magnetoelasticity (B_1) and magnetostriction (λ_{100}) that are comparable to previous reports of both bulk² and thin film samples¹ at similar concentrations (Fig. R1a and again in R2). The reproducibility is a validation of our approach. The reviewer presents an interesting idea, but we have not attempted to anneal our samples away from the A2 phase. As part of our experiments, we observed that our samples were thermally stable up to ~300C. One challenge will be to anneal at elevated temperatures without the substrate losing oxygen to the FeGa layer. One potential solution might be to integrate an epitaxial diffusion barrier such as MgO. We thank the reviewer for the idea and may explore this in a future work.

3) Did the authors give a look to the [113] diffraction peaks that are typical of the D03 structure?

We agree that both 113 and 001 are chemically sensitive diffraction peaks. Here, the 001 peak discerns the A2 phase from the B2/D03 phases while the 113 peak distinguishes D03 vs. A2/B2. In **Sup. Note 1**, we have added a detailed calculation of structure factor and chemical sensitivities at 113 and 001. Per the structure factors, the lack of a 001 diffraction peak means we are in a pure A2 phase, as the 001 can come from either the B2 or D03 phases. As the 113 peak can only come from the D03 phase, a missing 113 would mean we would be in some combination of the A2 and B2 phases. Further the 113 and 001 diffraction peaks have the same intensity based on structure factor with 001 being the lower index, and therefore lower Bragg angle, the 001 peak will be easier to detect in practice. Since our TEM and XRD measurements show no evidence of a 001 diffraction peak, we believe this is strong evidence for the absence of secondary phases.

Supplementary Note 1 | Structure factor of FeGa phases.

The A2, B2 and D03 structures can be represented in terms of the base structure (L2₁) with three atomic sites (A, B, C). For the solid solution A2, all three sites are equal. For B2, the B and C sites are equal, but distinct from the A site. Similarly, for D03, A and C are equal. The structure factor ($S(hkl)$) determines intensity of diffraction peaks at index (hkl). $S(hkl)$ for the base structure is:

$$S_{L2_1}(hkl) = \sum_i f_i \exp[-i\mathbf{b}_{hkl} \cdot \mathbf{r}_i]$$

$$= f_A [1 + e^{-i\pi h} + e^{-i\pi k} + e^{-i\pi l} + e^{-i\pi(h+k)} + e^{-i\pi(k+l)} + e^{-i\pi(h+l)} + e^{-i\pi(h+k+l)}]$$

$$+ f_B \left[e^{-\frac{i\pi}{2}(h+k+l)} + e^{-\frac{i\pi}{2}(3h+3k+l)} + e^{-\frac{i\pi}{2}(h+3k+3l)} + e^{-\frac{i\pi}{2}(3h+k+3l)} \right]$$

$$+ f_C \left[e^{-\frac{i\pi}{2}(h+k+3l)} + e^{-\frac{i\pi}{2}(3h+3k+3l)} + e^{-\frac{i\pi}{2}(h+3k+l)} + e^{-\frac{i\pi}{2}(3h+k+l)} \right],$$

where \mathbf{b}_{hkl} , \mathbf{r}_i , f_i denote the reciprocal lattice vector at (hkl), basis vector and scattering factor respectively.

The [002] peak (equivalent to [001] peak for A2) is chemically sensitive to the A2 structure, as a peak can arise from either the B2 or D03 phases:

$$S_{L2_1}(002) = 4[2f_A - (f_B + f_C)]$$

$$S_{A2}(002) = 0$$

$$S_{B2}(002) = 8(f_A - f_B)$$

$$S_{D03}(002) = 4(f_A - f_B)$$

Similarly [113] peak is chemically sensitive to D03, as the peak can only arise from the D03 phase:

$$S_{L2_1}(113) = 4i(f_C - f_B)$$

$$S_{A2}(113) = 0$$

$$S_{B2}(113) = 0$$

$$S_{D03}(113) = 4i(f_A - f_B)$$

Therefore, the absence of a [002] for peak in SAED (**Fig. 1d**) is a good indicator of the pure A2 phase. Note that the [002] peak is equivalent to $[001]_{A2,B2}$, because A2 and B2 have a smaller conventional unit cell.

Minor concerns:

1) which is the thickness of the substrates?

PMN-PT substrates are 1mm thick in all cases. This is mentioned in the methods but has been added to the main text for clarity (pg 3 line 26). The text now reads:

“Our samples consist of epitaxial (001)-oriented, 15 nm thick A2 (α -Fe) phase, magnetostrictive $\text{Fe}_{1-x}\text{Ga}_x$ ($x = 0.215, 0.245, 0.30$) single-crystal films on 1 mm-thick (001)-oriented PMN-PT substrates...”

2) Authors say that the D03 structure is bcc. I'd rather say that it is bcc-like.

This has been changed in the text (pg 3 line 2). The text now reads:

“ $\text{Fe}_{1-x}\text{Ga}_x$ undergoes a phase transition at ~18% Ga from a disordered A2 phase, at low concentrations of Ga, to an ordered BCC-like phase (D0_3)...”

3) Did the authors try to follow magnetization by Kerr when the voltage is applied on the magnetoelectric device? The authors suppose that the system is single domain when they evaluate α_{eff} , don't they?

MOKE was performed on preliminary samples, but for such a thin metal layer birefringence from the ferroelectric is a significant concern and thus was not done for the remainder.

The referee correctly points out that magnetoresistance measurements of the bar probe the composite magnetic domain state from the whole device. Samples are switched electrically, then magnetized with a field along the device direction to set the magnetization into a single domain state. The magnetic field is then reduced for the low field AMR, which shows the direction of magnetization as the phase shift, ϕ , in $R = \cos(2\theta + \phi)$. The electric field is then stepped, allowed to settle, and the process is repeated from 400 V \rightarrow -400 V \rightarrow 400 V in 50 V increments.

Figure R4 | Sample AMR data. Data showing angle resolved low-field magnetoresistance of the FeGa devices, where the phase shift corresponds to the direction of magnetization, demonstrating that they behave as a single magnetic domain with one orientation, not as a collection of magnets.

4) What is the estimated error bar on α_{eff} ?

The sources of error in the measurement are the angular resolution of the AMR, the variance of fitting the data to $\cos(2\theta)$, and the resolution in electric field. Angular data are taken in 10° increments, corresponding to a 5° uncertainty in the magnetization angle. Electric field data are taken in 1 or 0.5 kV/cm increments, corresponding to a 0.5 or 0.25 kV/cm uncertainty. Magnetoelectric hysteresis and magnetoelectric coefficients in **Sup. Figure S3** have been updated accordingly.

Supplementary Figure S3 | Complete magnetoelectric switching data. Example data taken at an electrical bias of up to ± 400 V across the sample, used to reconstruct the hysteresis in magnetization direction and converse magnetoelectric coefficient. AMR curves taken as a function of field angle are fit to $\cos(2\theta)$ where the phase shift corresponds to the direction of magnetization, ϕ . The two saturated

polarization states of the ferroelectric show a 90° phase shift in the curve, demonstrating a 90° switching of magnetization. The error bars represent the $\pm 5^\circ$ variance from the angular resolution, the 1.0/0.5 kV cm^{-1} resolution in electric field, and one standard deviation of the fit to $\cos(2\theta)$. This measurement is done by poling the magnetization along the easy axis and using low-field directional AMR to probe the rotation of the uniaxial, strain-induced anisotropy. Thus, the apparent handedness is a result of the direction used in easy-axis poling and not indicative of the device itself, but is a direct probe of the anisotropy direction.

Figure 2c has also been updated with these same data.

5) I did not understand how Figure 3 d is obtained. It's not clear for me, sorry.

The text in the methods has been changed to the following to make this procedure clearer:

“To construct the composite switching maps, the sample is first poled at +400 V and PFM is done. The combination of in-plane and out-of-plane PFM responses can be used to map the direction of polarization in 3D space. The sample is then poled at -100/-200/-400 V and the same PFM mapping is done. The two images, +400 V and -100/-200/-400 V, are then compared pixel-by-pixel to create a map of the switching events where the color corresponds to the angle between the polarization vectors of before and after images. E.g. if a pixel maps to a (+x,+y,+z) polarization after +400 V, then a (+x,+y,-z) polarization after -400 V, that corresponds to a 71° out-of-plane switching event which is shown as yellow in **Figure 3d** and **Sup. Figure S5**.”

References

1. Barturen, M. *et al.* Bulklike behavior of magnetoelasticity in epitaxial $\text{Fe}_{1-x}\text{Ga}_x$ thin films. *Phys. Rev. B* **99**, 134432 (2019).
2. Clark, A. E. *et al.* Extraordinary magnetoelasticity and lattice softening in bcc Fe-Ga alloys. *Journal of Applied Physics* **93**, 8621–8623 (2003).
3. Clark, A. E. *et al.* Temperature dependence of the magnetic anisotropy and magnetostriction of $\text{Fe}_{100-x}\text{Ga}_x$ ($x=8.6, 16.6, 28.5$). *Journal of Applied Physics* **97**, 10M316 (2005).
4. Petculescu, G., Hathaway, K. B., Lograsso, T. A., Wun-Fogle, M. & Clark, A. E. Magnetic field dependence of galfenol elastic properties. *Journal of Applied Physics* **97**, 10M315 (2005).
5. Yang, L. *et al.* Bipolar loop-like non-volatile strain in the (001)-oriented $\text{Pb}(\text{Mg}_{1/3}\text{Nb}_{2/3})\text{O}_3$ - PbTiO_3 single crystals. *Scientific Reports* **4**, 4591 (2014).
6. Guo, X. *et al.* Electrical field control of non-volatile 90° magnetization switching in epitaxial FeSi films on (001) $0.7[\text{Pb}(\text{Mg}_{1/3}\text{Nb}_{2/3})\text{O}_3]-0.3[\text{PbTiO}_3]$. *Appl. Phys. Lett.* **108**, 042403 (2016).
7. Zhang, S. *et al.* Electric-Field Control of Nonvolatile Magnetization in $\text{Co}_{40}\text{Fe}_{40}\text{B}_{20}/\text{Pb}(\text{Mg}_{1/3}\text{Nb}_{2/3})_{0.7}\text{Ti}_{0.3}\text{O}_3$ Structure at Room Temperature. *Phys. Rev. Lett.* **108**, 137203 (2012).
8. Wang, H. *et al.* Understanding strong magnetostriction in Fe 100-x Ga x alloys. *Sci Rep* **3**, 1–5 (2013).
9. Wu, R. Origin of large magnetostriction in FeGa alloys. *Journal of Applied Physics* **91**, 7358–7360 (2002).

Reviewers' Comments:

Reviewer #2:

Remarks to the Author:

The authors response to my concerns on measure the magnetostriction of these thin films using direct methods. I understand their explanation about the difficulties of measuring directly the magnetostriction of these thin films. However, I think that such a discussion about the difficulties of measuring directly the magnetostriction of these thin films and the validation of their approach should be included in the text. Furthermore, the authors should weaken their statement about high magnetostriction performance. It is important to remind readers clearly that the high magnetostriction performance achieved in this work is obtained by an indirect measurement approach. I recommend minor revisions.

Reviewer #3:

Remarks to the Author:

The authors improved the quality of the article in a very short time. Still, I have many concerns that I summarize below.

In their response to referee, the authors compare their results with articles already available in literature, in particular Parkes et al. Scientific Reports 3, Article number: 2220 (2013) and Barturen et al. PRB 99, 134432 (2019). The authors notice that their results are corroborated by Parkes measurements for $x=19.5\%$. This should be reported in the text and the Parkes's paper should be cited since relevant for the article discussion.

On the other hand, the Barturen's paper present results that go up 25%. Authors say that Barturen's samples present a DO3 structure leading to a lowering of the magnetoelastic coupling. This is not true. In the Barturen's article it's clearly written that the DO3 phase is not detected. Moreover, Barturen et al. show that the magnetoelastic coefficients are not affected by the B2 or A2 structure. Of course, FeGa thin films grown on PZT may present different behaviours since growth parameters are not identical. By the way, I consider fair and honest to compare these new results on PZT with the Barturen's ones on GaAs, even if the results are not 'in phase' with. I also consider that the authors should clearly underline in the article that their evaluation of the magnetostrictive properties are clearly dependent on the elastic coefficients they used and that that they did not measure them in their own samples. They adopted bulk values from other samples given in literature (presenting the DO3 phase). They present DFT calculation that corroborate the Clark's measurements but that are not usefull for a correct evaluation of the elastic constants. This is a weak point of their analysis that should be much more underlined in the text. I believe that a Nature article would demand an elastic constant evaluation of the samples since the softening of the elastic constants determines the enhancement of the magnetostrictive coefficients in a dramatic way.

Similarly, the uncertainty associated with the PZT switching fraction is adopted from literature and not directly measured on their own samples. Once again, I believe that a Nature article warrants a direct measurement of the samples rather than an indirect, literature dependent, evaluation.

Concerning X ray data, I'd prefer that the author show a diffraction diagram of a high concentration sample rather than the 18% one where the DO3 phase is not expected. They should also give the lattice parameters of the measured samples since they depend strongly on the Ga-content (an indirect way to corroborate the Ga content estimation).

Still, I believe that the conclusions would be much more robust if the samples were annealed in order to modify the atomic structure. This doesn't take a great effort. In the paper by Eddrief et al. Phys. Rev. B 73, 115315 (2006)), it is shown that by annealing at very moderate temperatures (300°C) the crystal structure can be modified.

In conclusion, this work MAYBE indicates that FeGa grown on PZT present extraordinary magnetostrictive properties. By the way, a robust demonstration (as Nature articles demand) should require extra measurements on their own samples: the elastic constants and effective PZT

strain.

REVIEWER COMMENTS

We appreciate the overall positive response and constructive feedback from the reviewers and thank them for their comments. We have satisfied referees 1 and 2 and all have noted the novelty and impact of the work. Here, we have addressed the referee comments which are detailed below.

Reviewer #2 (Remarks to the Author):

The authors response to my concerns on measure the magnetostriction of these thin films using direct methods. I understand their explanation about the difficulties of measuring directly the magnetostriction of these thin films. However, I think that such a discussion about the difficulties of measuring directly the magnetostriction of these thin films and the validation of their approach should be included in the text. Furthermore, the authors should weaken their statement about high magnetostriction performance. It is important to remind readers clearly that the high magnetostriction performance achieved in this work is obtained by an indirect measurement approach. I recommend minor revisions.

For clarity, we have added the following sentence at the first mention of a quantitative magnetostriction value, pg. 3 line 13:

“Here, we investigate thin films of the magnetostrictive alloy $\text{Fe}_{1-x}\text{Ga}_x$ due to its earth abundance, high magnetostriction, and relatively unexplored phase space.

In the thin film regime, direct measurement of mechanical properties, such as magnetostriction, is difficult and remains an active area of research. State-of-the-art techniques can evaluate elastic moduli of simple systems with thicknesses on the order of multiple 10s of nm, often by borrowing assumptions from bulk materials¹⁻⁴. In magnetics, magnetostriction of thin films may be extracted indirectly from their magnetoelastic coupling coefficients⁵ by analyzing strain-induced change in magnetic anisotropy^{6,7}. Our approach takes direction from these existing methodologies⁵⁻⁷ but leveraging strain from an intrinsic piezoelectric component. By evaluating the magnetostriction of $\text{Fe}_{1-x}\text{Ga}_x$ through magnetoelectric measurements and theoretical analysis based on piezoelectric domain imaging, we present a means to boost the magnetostriction by as much as 20x...”

We have also extended the existing statement and added the following text to pg. 5 line 19:

“Magnetostriction has been measured by laser interferometry under a magnetic field⁸, but the in-plane magnetization and strain of our samples largely invalidates this technique. Additionally, clamping of the film to the substrate prevents the use of a strain

gauge under magnetic field, as the substrate should quench any response. The magnetostriction of thin films is typically extracted indirectly from their magnetoelastic coupling coefficients⁵ because a direct measure of the static magnetostrictive coefficient is difficult in a thin film geometry due to substrate interaction.”

And to pg. 8 line 1 to make the mechanism and the significant difference from existing results clearer:

“In bulk $\text{Fe}_{1-x}\text{Ga}_x$, the pre-factor $(c_{11} - c_{12})/2$ can range from 28 GPa to 7 GPa due to the variation of the Ga composition^{9,10} (from 17% to 27.2%). $(c_{11} - c_{12})/2$ values from DFT simulation of disordered $\text{Fe}_{1-x}\text{Ga}_x$ agree with published experimental bulk values from ref. ⁹ to within the error bar of the simulation (**Figure 4a**), justifying their use here for our thin film samples. Additionally, these bulk values appear to be constant between phases, as $(c_{11} - c_{12})/2$ follows a linear trend across the 19% Ga threshold, and absent of an anomaly that would clearly correspond to the phase change in bulk. This reveals that the nature of the magnetostrictive transition is electronic and manifests only in B1. Using these values for $\varepsilon_{22} - \varepsilon_{11}$ and $(c_{11} - c_{12})/2$, we calculate a λ_{100} in the system between ranging between 300 ppm (21.5% Ga) and 3500 ppm (30% Ga sample), significantly higher than reported bulk values⁹ ($\lambda_{100} \cong 150$ ppm). Comparatively, previously published thin film results^{8,11} indicate a bulk-like behavior in FeGa grown epitaxially on GaAs substrates. Compositions studied near the phase boundary at 19%Ga show relative agreement between measured magnetoelastic (B_1) coefficients. Samples studied here, however, deviate significantly from bulk behavior after 19% and the large discrepancy appears to be due to the magnetoelastic (B_1) coefficient which, in these previous reports, shows a sharp drop after the 19% Ga threshold (**Figure 4b**). We do not observe this phenomenon in our samples, which leads to the large enhancement in magnetostriction considering bulk mechanical data, which follows a linear trend that is irrespective of phase¹². This enhancement of the magnetoelasticity, and thus magnetostriction...”

Figure 4 has also been updated to be more explicit about the underlying values and calculation.

Figure 4 | Enhanced magnetostriction coefficient through epitaxial stabilization. **a**, Plot of mechanical coefficient $\frac{1}{2}(c_{11} - c_{12})$ extracted from literature (red) and simulated here with DFT (blue). Both data sets follow approximately the same trend and show no deviation from linear behavior following the 19% phase limit. Literature values are from ref. ¹². **b**, Plot of magnetoelastic coefficients (B_1) taken from previous thin film (red) and bulk (grey) works compared to our measured values. We note that in previous works, there is a sharp decline in B_1 following the phase change at 19%Ga, which we do not observe. Bulk values (grey) are from refs. ^{9,13} and film values are from ref. ⁸. **c**, The extracted magnetostriction values as a function of Ga concentration with our values (red, open circles) compared to the measured bulk coefficients (blue) from ref. ⁹. The values from this work are plotted as $\frac{3}{2}\lambda_{100}$ to facilitate comparison with the bulk, polycrystalline values. Above 19% Ga, we do not observe a decrease in the magnetostriction associated with the formation of the ordered D0_3 phase and we extend the regime of the disordered A2 phase via epitaxial stabilization. As the concentration approaches the second phase change at $\sim 30\%$ Ga, the shear modulus $c_{11} - c_{12}$ approaches 0, leading to extremely large values of the magnetostriction. Calculation of the error bars, σ_λ , is detailed in Sup. Note 4. **d**, Comparison of the magnetostriction coefficients from this work to other magnetostrictive materials. The largest magnetostrictive tensor component λ_{ijk} of each respective material is plotted here for ease of comparison. Comparative data in **b** from refs. ^{14–22}.

Reviewer #3 (Remarks to the Author):

The authors improved the quality of the article in a very short time. Still, I have many concerns that I summarize below.

In their response to referee, the authors compare their results with articles already available in literature, in particular Parkes et al. Scientific Reports 3, Article number: 2220

(2013) and Barturen et al. PRB 99, 134432 (2019). The authors notice that their results are corroborated by Parkes measurements for $x=19.5\%$. This should be reported in the text and the Parkes's paper should be cited since relevant for the article discussion. On the other hand, the Barturen's paper present results that go up 25%. Authors say that Barturen's samples present a DO3 structure leading to a lowering of the magnetoelastic coupling. This is not true. In the Barturen's article it's clearly written that the DO3 phase is not detected. Moreover, Barturen et al. show that the magnetoelastic coefficients are not affected by the B2 or A2 structure. Of course, FeGa thin films grown on PZT may present different behaviours since growth parameters are not identical. By the way, I consider fair and honest to compare these new results on PZT with the Barturen's ones on GaAs, even if the results are not 'in phase' with.

I also consider that the authors should clearly underline in the article that their evaluation of the magnetostrictive properties are clearly dependent on the elastic coefficients they used and that that they did not measure them in their own samples. They adopted bulk values from other samples given in literature (presenting the DO3 phase). They present DFT calculation that corroborate the Clark's measurements but that are not useful for a correct evaluation of the elastic constants. This is a weak point of their analysis that should be much more underlined in the text. I believe that a Nature article would demand an elastic constant evaluation of the samples since the softening of the elastic constants determines the enhancement of the magnetostrictive coefficients in a dramatic way.

In order to make the challenge of mechanical measurements in ultrathin films more explicit, we have added the following text on pg. 3 line 13:

“Here, we investigate thin films of the magnetostrictive alloy $\text{Fe}_{1-x}\text{Ga}_x$ due to its earth abundance, high magnetostriction, and relatively unexplored phase space.

In the thin film regime, direct measurement of mechanical properties, such as magnetostriction, is difficult and remains an active area of research. State-of-the-art techniques can evaluate elastic moduli of simple systems with thicknesses on the order of multiple 10s of nm, often by borrowing assumptions from bulk materials¹⁻⁴. In magnetics, magnetostriction of thin films may be extracted indirectly from their magnetoelastic coupling coefficients⁵ by analyzing strain-induced change in magnetic anisotropy^{6,7}. Our approach takes direction from these existing methodologies⁵⁻⁷ but leveraging strain from an intrinsic piezoelectric component. By evaluating the magnetostriction of $\text{Fe}_{1-x}\text{Ga}_x$ through magnetoelectric measurements and theoretical analysis based on piezoelectric domain imaging, we present a means to boost the magnetostriction by as much as 20x...”

As stated in the text on pg. 5 line 28:

“Magnetostriction has been measured by laser interferometry under a magnetic field ⁸, but the in-plane magnetization and strain of our samples largely invalidates this technique. Additionally, clamping of the film to the substrate prevents the use of a strain gauge under magnetic field, as the substrate should quench any response. The magnetostriction of thin films is typically extracted indirectly from their magnetoelastic coupling coefficients⁵ because a direct measure of the static magnetostrictive coefficient is difficult in a thin film geometry due to substrate interaction. Coupling coefficients are normally determined by analyzing strain-induced change in magnetic anisotropy (or easy-axis reorientation) using analytical models^{6,7}. Our approach of determining magnetostriction is similar to these well-established methods⁵⁻⁷, except that strain is applied dynamically via the piezoelectric layer...”

We have added the Clark reference, modified the text to pg. 8 line 3, and added the elastic data to figure 4 to make the mechanism and the significant difference from existing results clearer:

“In bulk $\text{Fe}_{1-x}\text{Ga}_x$, the pre-factor $(c_{11} - c_{12})/2$ can range from 28 GPa to 7 GPa due to the variation of the Ga composition^{9,10} (from 17% to 27.2%). $(c_{11} - c_{12})/2$ values from DFT simulation of disordered $\text{Fe}_{1-x}\text{Ga}_x$ agree with published experimental bulk values from ref. ⁹ to within the error bar of the simulation (**Figure 4a**), justifying their use here for our thin film samples. Additionally, these bulk values appear to be constant between phases, as $(c_{11} - c_{12})/2$ follows a linear trend across the 19% Ga threshold, and absent of an anomaly that would clearly correspond to the phase change in bulk. This reveals that the nature of the magnetostrictive transition is electronic and manifests only in B1. Using these values for $\varepsilon_{22} - \varepsilon_{11}$ and $(c_{11} - c_{12})/2$, we calculate a λ_{100} in the system between ranging between 300 ppm (21.5% Ga) and 3500 ppm (30% Ga sample), significantly higher than reported bulk values⁹ ($\lambda_{100} \cong 150$ ppm). Comparatively, previously published thin film results^{8,11} indicate a bulk-like behavior in FeGa grown epitaxially on GaAs substrates. Compositions studied near the phase boundary at 19%Ga show relative agreement between measured magnetoelastic (B_1) coefficients. Samples studied here, however, deviate significantly from bulk behavior after 19% and the large discrepancy appears to be due to the magnetoelastic (B_1) coefficient which, in these previous reports, shows a sharp drop after the 19% Ga threshold (**Figure 4b**). We do not observe this phenomenon in our samples, which leads to the large enhancement in magnetostriction considering bulk mechanical data, which follows a linear trend that is irrespective of phase¹². This enhancement of the magnetoelasticity, and thus magnetostriction...”

We have also updated Figure 4 to show these existing results and highlight the differences here.

Figure 4 | Enhanced magnetostriction coefficient through epitaxial stabilization. **a**, Plot of mechanical coefficient $\frac{1}{2}(c_{11} - c_{12})$ extracted from literature (red) and simulated here with DFT (blue). Both data sets follow approximately the same trend and show no deviation from linear behavior following the 19% phase limit. Literature values are from ref. ¹². **b**, Plot of magnetoelastic coefficients (B_1) taken from previous thin film (red) and bulk (grey) works compared to our measured values. We note that in previous works, there is a sharp decline in B_1 following the phase change at 19%Ga, which we do not observe. Bulk values (grey) are from refs. ^{9,13} and film values are from ref. ⁸. **c**, The extracted magnetostriction values as a function of Ga concentration with our values (red, open circles) compared to the measured bulk coefficients (blue) from ref. ⁹. The values from this work are plotted as $\frac{3}{2}\lambda_{100}$ to facilitate comparison with the bulk, polycrystalline values. Above 19% Ga, we do not observe a decrease in the magnetostriction associated with the formation of the ordered $D0_3$ phase and we extend the regime of the disordered A2 phase via epitaxial stabilization. As the concentration approaches the second phase change at ~30% Ga, the shear modulus $c_{11} - c_{12}$ approaches 0, leading to extremely large values of the magnetostriction. Calculation of the error bars, σ_λ , is detailed in Sup. Note 4. **d**, Comparison of the magnetostriction coefficients from this work to other magnetostrictive materials. The largest magnetostrictive tensor component λ_{ijk} of each respective material is plotted here for ease of comparison. Comparative data in **b** from refs. ^{14–22}.

While it is true that we do not measure the elastic constants of the film directly, direct measurement of mechanical properties in ultrathin films is a significant challenge for the thin film community and is not yet possible on the scale of our films. In our calculations of the mechanical coefficients, we see good agreement with the bulk values and trend. DFT calculations are carried out on a volumetrically constrained supercell, to better mimic the conditions of the thin film, and the agreement here strongly implies that the material behaves mechanically similar, regardless of the phase and the boundary conditions. Additionally, in bulk, there is no change in the slope across the phase A2- $D0_3$ phase change, which further implies that the two phases behave very similarly mechanically.

From our analysis, the primary difference between the thin film and bulk behavior at high Ga concentrations comes from the B_1 coefficient, which drops at 19% in bulk. We do not observe this in our measurements.

Similarly, the uncertainty associated with the PZT switching fraction is adopted from literature and not directly measured on their own samples. Once again, I believe that a Nature article warrants a direct measurement of the samples rather than an indirect, literature dependent, evaluation.

We have expanded our PFM data set with an additional 20 scans (10 +4 V to -4 V switching events). The reported switching fraction of our PMN-PT layer is based on statistics of our own data set. The 109° switching fraction is $\sim 23\% \pm 4\%$. This error does not include measurements taken from literature. PMN-PT is a well-documented ferroelectric piezoelectric and our values agree with prior literature. This statement is updated in the text on pg. 7, line 21:

“Here, we experimentally measure this fraction of 109° switching by comparing composite piezoelectric force microscopy (PFM) micrographs at fields both before and after the magnetoelectric switching event, an example of which is shown in **Figure 3d** (full data set shown in **Sup. Figure 5**). From repetition of this measurement (**Figure 3e**), we estimate that $\eta_{109} \cong 23\% \pm 4\%$, consistent with published values...”

Figure 3 and **Sup. Figure 5** have been updated to show these additional switching maps and statistics. The calculations in **Figure 4** have also been updated to show this additional error

Figure 3 | Local shear strains arising from 109° polarization switching in PMN-PT. a, Polarizations within the $(110)_s$ (substrate) plane, blue, and the $(\bar{1}10)_s$ plane, green, are associated with shear distortion

in the $(001)_{pc}$ plane, **b**, indicated by the blue and green dashed frame. The corresponding shear strain arising from a 109° polarization switching can be calculated based on the coordinates of points r_i ($i=1,2,3,4$), where the translation from r_1 to r_2 results in a 0.192% shear strain per unit cell. This is then scaled by the fraction of ferroelectric domains that undergo a 109° switch (η_{109°) to calculate the total strain seen by the device. **c**, PFM switching map that allows us to experimentally determine η_{109° . This map is made by overlaying PFM micrographs before switching ($+4 \text{ kV cm}^{-1}$) and after switching (-4 kV cm^{-1}) and calculating the 3D switching angle per pixel. The directions of the ferroelectric vectors were determined by combining in-plane and out-of-plane piezoresponse patterns before and after rotating the sample by 90° to allow for the determination of in-plane directionality. The full data set is shown in **Supplementary Figure S5**. **d**, Histogram of the switching events from 14 composite images, with standard deviations shown as error bars. The analysis indicates that $23\% \pm 4\%$ of the domains undergo 109° switching.

Supplementary Figure S5 | Domain fraction of ferroelectric switching. PFM switching map that allows us to experimentally determine η_{109° . This map is made by overlaying PFM micrographs before switching (+4 kV cm⁻¹) and after switching -4 kV cm⁻¹) and calculating the 3D switching angle per pixel. The directions of the ferroelectric vectors were determined by combining in-plane and out-of-plane piezoresponse patterns before and after rotating the sample by 90° to allow for the determination of in-plane directionality.

Concerning X ray data, I'd prefer that the author show a diffraction diagram of a high concentration sample rather than the 18% one where the DO3 phase is not expected. They should also give the lattice parameters of the measured samples since they depend strongly on the Ga-content (an indirect way to corroborate the Ga content estimation).

Because the films are so thin (15 nm), XRD is not a good method to determine the phase fraction due to the sensitivity. As stated by the reviewer in their first review, diffraction at this scale only "...indicates that the B2 and the D03 phases, if they exist, are not well ordered. It's very likely that nanoclusters with Ga pairs are present in the samples, as discussed in ref.17." This situation is ideal for a local probe of structure, such as STEM, in which we see no evidence of other phases.

Sup. Figure 1 has been amended to show RHEED patterns of films during deposition to provide a more accurate picture of bulk diffraction, as well as electron diffraction data from all compositions.

Supplementary Figure S1 | Diffraction data from FeGa. **a**, In-situ reflection high energy electron diffraction (RHEED) data on the $[110]_s$ azimuth of the PMN-PT substrate and the $[100]$ of the FeGa film. RHEED data shows no signal corresponding to the 100 peak of the FeGa, indicating that the films are in the disordered, A2-like phase. **b**, Select area electron diffraction (SAED) pattern of FeGa thin films, which confirms FeGa thin film is in the A2 phase rather than intermetallic B2 or D03 across different Ga concentrations. FeGa Bragg peaks FeGa (yellow dashed line) is extinct when sum of reciprocal lattice indices are odd, which occurs only in the A2 phase as detailed below in **Sup Note 1**. The out-of-plane lattice constants of the FeGa film calculated from diffraction images are shown with error bars.

The compositions of our samples are not estimated but are measured directly using EELS and RBS. The out-of-plane (c -axis) lattice constants are calculated above from diffraction data, but because the error bars are so significant, a trend cannot be reasonably determined.

Still, I believe that the conclusions would be much more robust if the samples were annealed in order to modify the atomic structure. This doesn't take a great effort. In the paper by Eddrief et al. Phys. Rev. B 73, 115315 (2006)), it is shown that by annealing at very moderate temperatures (300°C) the crystal structure can be modified.

The article²³ cited above titled "Interface bonding of a ferromagnetic/semiconductor junction: A photoemission study of Fe/ZnSe(001)" discusses photoemission studies of a Fe/ZnSe interface and does not discuss annealing or have relevance to our study.

Attempting to anneal our samples away from the A2 phase, we observe degradation of the heterostructure before any structural change in the FeGa. Annealing the sample in steps of 50 °C for 5 hours in vacuum, we see no change in the magnetic properties of the film, which should telegraph any structural change, up to ~650 °C. At this temperature, we still observe no change in the crystal structure of the FeGa film, shown by SAED in **Figure R5a**, but the surface of the PMN-PT begins to degrade, ruining the functionality of the heterostructure. With films of this thickness in this heterostructure, annealing appears to be insufficient to drive a phase change away from the A2 phase.

Figure R5 | Thermal stability of FeGa samples. **a** SAED pattern of a $\text{Fe}_{70}\text{Ga}_{30}$ after annealing for 5 hours in vacuum at 700 °C. The FeGa film remains in the disordered A2 phase, indicated by absence of the ordered 001 peak. In ADF-STEM, **b**, we observe a degradation of the surface of the PMN-PT, which forms a ~30 nm amorphous layer beneath the FeGa. This amorphous layer serves to destroy the functionality of

the sample. The apparent periodicity in crystalline PMN-PT is an aliasing artifact and not representative of the crystal orientation.

In conclusion, this work MAYBE indicates that FeGa grown on PZT present extraordinary magnetostrictive properties. By the way, a robust demonstration (as Nature articles demand) should require extra measurements on their own samples: the elastic constants and effective PZT strain.

We have completely addressed the reviewer's comments from the first review concerning rigorous calculation of error in our results and phase evaluation. We have also addressed the reviewers new comments and made our calculations more transparent and demonstrated that they are significantly distinct from existing results^{8,11} in terms of composite structure, composition, and mechanism and we have significantly increased the amount of data used in the calculation of the ferroelectric strain.

As STEM is the most sensitive technique used to determine crystalline structure at these length scales, the data unambiguously show that the samples are in the disordered phase. We confirm this mechanism through direct observation of the concentration using RBS and EELS, and show using diffraction that the A2 phase is invariant even at these high Ga concentrations.

Our approach builds on existing standards for the evaluation of magnetostriction but does it in a way that uses the dynamic operation of the device. With this technique, we can both reproduce existing results for low-Ga-composition films^{8,9,11,12}, as well as evaluate our novel, metastable FeGa composition with high magnetostriction.

This article reflects Nature's standards of rigor and impact.

References:

1. Kopycinska-Müller, M., Geiss, R. H., Müller, J. & Hurley, D. C. Elastic-property measurements of ultrathin films using atomic force acoustic microscopy. *Nanotechnology* **16**, 703–709 (2005).
2. Hernandez-Charpak, J. N. *et al.* Full Characterization of the Mechanical Properties of 11–50 nm Ultrathin Films: Influence of Network Connectivity on the Poisson's Ratio. *Nano Lett.* **17**, 2178–2183 (2017).
3. Emori, S. *et al.* Coexistence of Low Damping and Strong Magnetoelastic Coupling in Epitaxial Spinel Ferrite Thin Films. *Advanced Materials* **29**, 1701130 (2017).
4. Zhang, Y., Wang, H., Li, X., Tang, H. & Polycarpou, A. A. A finite element correction method for sub-20 nm nanoindentation considering tip bluntness. *International Journal of Solids and Structures* **129**, 49–60 (2017).
5. Song, O., Ballentine, C. A. & O'Handley, R. C. Giant surface magnetostriction in polycrystalline Ni and NiFe films. *Appl. Phys. Lett.* **64**, 2593–2595 (1994).

6. O'Handley, R. C., Song, O. & Ballentine, C. A. Determining thin-film magnetoelastic constants. *Journal of Applied Physics* **74**, 6302–6307 (1993).
7. Sander, D. The correlation between mechanical stress and magnetic anisotropy in ultrathin films. *Rep. Prog. Phys.* **62**, 809–858 (1999).
8. Barturen, M. *et al.* Bulklike behavior of magnetoelasticity in epitaxial $\text{Fe}_{1-x}\text{Ga}_x$ thin films. *Phys. Rev. B* **99**, 134432 (2019).
9. Clark, A. E. *et al.* Extraordinary magnetoelasticity and lattice softening in bcc Fe-Ga alloys. *Journal of Applied Physics* **93**, 8621–8623 (2003).
10. Datta, S., Atulasimha, J., Mudivarthi, C. & Flatau, A. B. Stress and magnetic field-dependent Young's modulus in single crystal iron–gallium alloys. *Journal of Magnetism and Magnetic Materials* **322**, 2135–2144 (2010).
11. Parkes, D. E. *et al.* Magnetostrictive thin films for microwave spintronics. *Scientific Reports* **3**, 2220 (2013).
12. Petculescu, G., Hathaway, K. B., Lograsso, T. A., Wun-Fogle, M. & Clark, A. E. Magnetic field dependence of galferol elastic properties. *Journal of Applied Physics* **97**, 10M315 (2005).
13. Clark, A. E. *et al.* Temperature dependence of the magnetic anisotropy and magnetostriction of $\text{Fe}_{100-x}\text{Ga}_x$ ($x=8.6, 16.6, 28.5$). *Journal of Applied Physics* **97**, 10M316 (2005).
14. Grössinger, R., Turtelli, R. S. & Mehmood, N. Materials with high magnetostriction. *IOP Conf. Ser.: Mater. Sci. Eng.* **60**, 012002 (2014).
15. Srinivasan, G. *et al.* Magnetolectric bilayer and multilayer structures of magnetostrictive and piezoelectric oxides. *Phys. Rev. B* **64**, 214408 (2001).
16. Lo, C. C. H., Ring, A. P., Snyder, J. E. & Jiles, D. C. Improvement of magnetomechanical properties of cobalt ferrite by magnetic annealing. *IEEE Transactions on Magnetics* **41**, 3676–3678 (2005).
17. Wang, D., Nordman, C., Qian, Z., Daughton, J. M. & Myers, J. Magnetostriction effect of amorphous CoFeB thin films and application in spin-dependent tunnel junctions. *Journal of Applied Physics* **97**, 10C906 (2005).
18. Özkale, B., Shamsudhin, N., Bugmann, T., Nelson, B. J. & Pané, S. Magnetostriction in electroplated CoFe alloys. *Electrochemistry Communications* **76**, 15–19 (2017).
19. Lou, J. *et al.* Soft magnetism, magnetostriction, and microwave properties of FeGaB thin films. *Appl. Phys. Lett.* **91**, 182504 (2007).
20. Muth, P. & Wohlfarth, E. P. *Ferromagnetic Materials, Volume 1*. vol. 16 (North-Holland Publishing Company, 1981).
21. Samata, H., Fujiwara, N., Nagata, Y., Uchida, T. & Der Lan, M. Magnetic anisotropy and magnetostriction of SmFe_2 crystal. *Journal of Magnetism and Magnetic Materials* **195**, 376–383 (1999).
22. Summers, E. M., Lograsso, T. A. & Wun-Fogle, M. Magnetostriction of binary and ternary Fe–Ga alloys. *J Mater Sci* **42**, 9582–9594 (2007).
23. Eddrief, M. *et al.* Interface bonding of a ferromagnetic/semiconductor junction: A photoemission study of $\text{Fe}/\text{ZnSe}(001)$. *Phys. Rev. B* **73**, 115315 (2006).

Reviewers' Comments:

Reviewer #2:

Remarks to the Author:

Since the manuscript has been improved greatly, I agree to accept as is.

Reviewer #3:

Remarks to the Author:

The authors improved a lot the quality of the manuscript and expanded data presentation. I believe that the authors should cite their attempts to anneal the sample since in the article entitled 'Metastable tetragonal structure of Fe_{100-x}Gax epitaxial thin films on ZnSe/GaAs(001) substrate' by M. Eddrief, et al. Phys. Rev. B 84, 161410(R), (2011) ` this method was proven to be fruitful to induce structural changes.

Furthermore, the comparison with Ref.28 should be modified: In Ref 28 the magnetization was in plane and Bi coefficients could be measured. The sentence `but the in-plane magnetization and strain of our samples largely invalidates this technique.' is not appropriate. I'd rather say that the thicknesses of the samples here are too thin to be evaluated by laser interferometry.

REVIEWERS' COMMENTS

Reviewer #2 (Remarks to the Author):

Since the manuscript has been improved greatly, I agree to accept as is.

We thank the reviewer for their positive comments.

Reviewer #3 (Remarks to the Author):

The authors improved a lot the quality of the manuscript and expanded data presentation. I believe that the authors should cite their attempts to anneal the sample since in the article entitled 'Metastable tetragonal structure of Fe_{100-x}Ga_x epitaxial thin films on ZnSe/GaAs(001) substrate' by M. Eddrief, et al. Phys. Rev. B 84, 161410(R), (2011) ' this method was proven to be fruitful to induce structural changes.

We thank the reviewer for their positive comments on our work.

The following text has been added to pg 4 line 23:

"In contrast to previous reports with FeGa thin films²⁸, attempting to anneal our samples away from the A2 phase, we observe degradation of the heterostructure before any structural change in the FeGa. Annealing the sample in vacuum, we see no change in the magnetic properties of the film, which should telegraph any structural change, up to ~650 °C. At this temperature, the surface of the PMN-PT begins to degrade, destroying the functionality of the heterostructure."

Furthermore, the comparison with Ref.28 should be modified: In Ref 28 the magnetization was in plane and Bi coefficients could be measured. The sentence 'but the in-plane magnetization and strain of our samples largely invalidates this technique.' is not appropriate. I'd rather say that the thicknesses of the samples here are too thin to be evaluated by laser interferometry.

The text on pg 6 line 7 has been modified to:

"In the past, magnetostriction has been measured by laser interferometry under a magnetic field ²⁸, but the thickness and chemistry of the substrate in our samples largely invalidates this technique."